# MePo: Meta Post-Refinement for Rehearsal-Free General Continual Learning

**Guanglong Sun** [1 2]  **Hongwei Yan** [1 2]  **Liyuan Wang** [3]  **Zhiqi Kang** [4]  **Shuang Cui** [5]
**Hang Su** [6]  **Jun Zhu** [6]  **Yi Zhong** [1 2]

## Abstract

To cope with uncertain changes of the external world, intelligent systems must continually learn from complex, evolving environments and respond in real time. This ability, collectively known as general continual learning (GCL), encapsulates practical challenges such as online datastreams and blurry task boundaries. Although leveraging pretrained models (PTMs) has greatly advanced conventional continual learning (CL), these methods remain limited in reconciling the diverse and temporally mixed information along a single pass, resulting in sub-optimal GCL performance. Inspired by meta-plasticity and reconstructive memory in neuroscience, we introduce here an innovative approach named **Me**ta **Po**st-Refinement (MePo) for PTMs-based GCL. This approach constructs pseudo task sequences from pretraining data and develops a bi-level meta-learning paradigm to refine the pretrained backbone, which serves as a prolonged pretraining phase but greatly facilitates rapid adaptation of representation learning to downstream GCL tasks. MePo further initializes a meta covariance matrix as the reference geometry of pretrained representation space, enabling GCL to exploit second-order statistics for robust output alignment. MePo serves as a plug-in strategy that achieves significant performance gains across a variety of GCL benchmarks and pretrained checkpoints in

a rehearsal-free manner (e.g., 15.10%, 13.36%, and 12.56% on CIFAR-100, ImageNet-R, and CUB-200 under Sup-21/1K). Our source code is available at MePo.

## 1. Introduction

Human learning is characterized by the remarkable adaptability to accumulate knowledge from complex, evolving environments and to respond in real time. While numerous efforts in continual learning (CL) (Wang et al., 2024; Van de Ven & Tolias, 2019) aim to construct AI models in a similar way, conventional settings have focused on offline learning of sequential tasks with disjoint task boundaries, which are out of touch with real-world scenarios. In this regard, the concept of general continual learning (GCL) (Buzzega et al., 2020; De Lange et al., 2021) has been proposed to cover a variety of practical challenges, particularly those with online datastreams and blurry task boundaries (Moon et al., 2023; Kang et al., 2025), making it increasingly difficult for AI models to rapidly capture and effectively balance successive information. Most existing methods that attempt GCL from scratch rely on replaying old training samples (Aljundi et al., 2019; Buzzega et al., 2020; Koh et al., 2021; Bang et al., 2021; Yan et al., 2024), which incurs additional memory costs and privacy risks. Without leveraging prior knowledge, these methods exhibit inferior learning efficacy, limited generalization capabilities, and severe catastrophic forgetting (Kang et al., 2025).

Recent advances in CL have shifted toward employing pretrained models (PTMs) and parameter-efficient tuning (PET) techniques for representation learning (Wang et al., 2022b;a), and recover old task distributions in representation space for output alignment (Zhang et al., 2023; McDonnell et al., 2024), which obtain superior performance in conventional CL settings in a rehearsal-free manner. Despite the promise, these methods still face significant challenges in GCL: mainstream PET techniques (e.g., visual prompt tuning (Yoo et al., 2023; Ma et al., 2023)) often fall short in capturing the nuances of online datastreams, while common strategies of approximating old task distributions rely on disjoint task boundaries. State-of-the-art GCL methods (Kang et al., 2025; Moon et al., 2023) perform contrastive regular-

---

[1]School of Life Sciences, IDG/McGovern Institute for Brain Research, Tsinghua University, Beijing, China [2]Tsinghua-Peking Center for Life Sciences [3]Department of Psychological and Cognitive Sciences, Tsinghua University, Beijing, China [4]Univ. Grenoble Alpes, Inria, CNRS, Grenoble INP, LJK, Grenoble, France [5]Institute of Software Chinese Academy of Sciences, Beijing, China [6]Dept. of Comp. Sci. and Tech., Institute for AI, Tsinghua-Bosch Joint ML Center, THBI Lab, BNRist Center, Tsinghua University, Beijing, China. Correspondence to: Liyuan Wang <liyuanwang@tsinghua.edu.cn>, Yi Zhong <zhongyithu@tsinghua.edu.cn>.

*Proceedings of the 43 [rd] International Conference on Machine Learning*, Seoul, South Korea. PMLR 306, 2026. Copyright 2026 by the author(s).

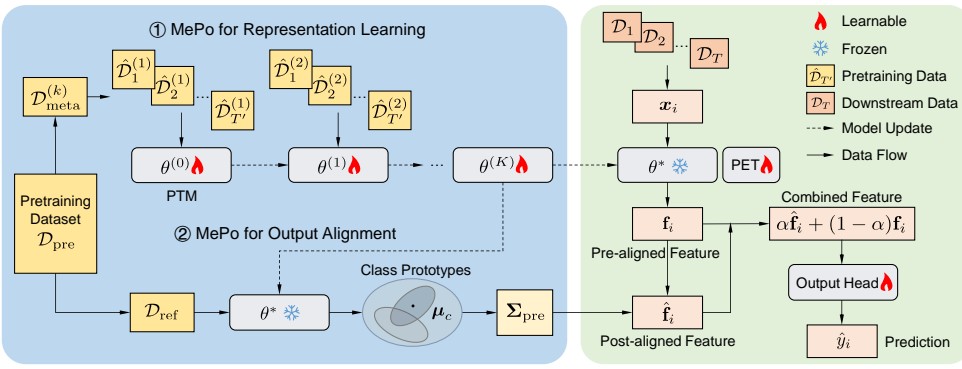

*Figure 1.* The proposed MePo framework for rehearsal-free general continual learning.

ization or initial session adaptation of prompt parameters, along with logit masking for balancing the output layer.[1] However, these methods fall short in fully addressing the two GCL challenges, especially under self-supervised PTMs that are more realistic yet often underfitted in their representations (see our empirical results in Sec. 2.2).

Compared to AI models, the biological brain enjoys strong GCL-like capabilities by imposing meta-plasticity (Abraham, 2008; Abraham & Bear, 1996; Sun et al., 2025) underlying the brain networks that retain substantial "pre-trained knowledge", positioning them in a critical state of neurodynamics for rapid adaptation. Up on meta-plasticity, the neural representations of incoming memories are continually encoded into and reconstructed from a shared representation space that enables real-time generalization, known as the reconstructive memory theory (Lei et al., 2022; 2024; Richards & Frankland, 2017). Inspired by such biological mechanisms, we propose an innovative approach named **Me**ta **Po**st-Refinement (MePo) for PTMs-based GCL (Fig. 1). MePo constructs pseudo tasks sequences from subsets of pretraining data, and develops a bi-level meta-learning paradigm to refine the pretrained backbone in a data-driven manner. This serves as a prolonged pretraining phase of *one-time cost*, but greatly facilitates rapid adaptation of representation learning to downstream GCL tasks *without additional overhead*. MePo further initializes a meta covariance matrix as the reference geometry of pretrained representation space, to which the features of incoming training samples are continually aligned and reconstructed, ensuring accurate and balanced predictions.

Unlike prior PTMs-based CL/GCL methods that rely solely on upstream pretraining or downstream adaptation, MePo extends the upstream pretraining with an additional post-refinement using pretraining data. To our knowledge, this is the first attempt that prepares PTMs for CL/GCL in advance, enabled by meta-learned pseudo task sequences for effective backbone refinement and meta-covariance for stable output

alignment. We perform extensive experiments to validate the proposed framework. MePo serves as a plug-in strategy that significantly improves recent strong PTMs-based CL and GCL methods across a variety of GCL benchmarks and pretrained checkpoints in a rehearsal-free manner (e.g., 15.10%, 13.36%, and 12.56% on CIFAR-100, ImageNet-R, and CUB-200 under Sup-21/1K), while ensuring resource efficiency during the GCL phase. Comprehensive ablation studies and visualization results confirm its adaptive benefits in both representation learning and output alignment.

**Declaration of interests:** The authors declare no competing interests.

## 2. Formulation and Preliminary Analysis

In this section, we first describe the problem setup of GCL, and then analyze the practical challenges of adapting state-of-the-art PTMs-based CL methods to GCL.

### 2.1. Problem Setup

Let's consider a neural network model comprising a backbone $f_\theta(\cdot)$ parameterized by $\theta$ and an output layer $h_\psi(\cdot)$ parameterized by $\psi$. The model needs to learn sequential tasks $t \in \{1, ..., T\}$ from their respective training sets $\mathcal{D}_1, ..., \mathcal{D}_T$. Each $\mathcal{D}_t$ consists of multiple data-label pairs $(\boldsymbol{x}_t, y_t)$, where the input data $\boldsymbol{x}_t \in \mathcal{X}_t$ and its ground-truth label $y_t \in \mathcal{Y}_t$ have respective spaces. For classification tasks, we further denote $|\mathcal{Y}_t|$ as the number of classes observed in task $t$. The objective of CL is to learn a mapping function from $\mathcal{X} = \bigcup_{t=1}^{T} \mathcal{X}_t$ to $\mathcal{Y} = \bigcup_{t=1}^{T} \mathcal{Y}_t$, so as to predict the label $\hat{y} = h_\psi(f_\theta(\boldsymbol{x}))$ of any unseen test data $\boldsymbol{x}$ belonging to the previous tasks.

In conventional CL settings, the task-wise input spaces (for DIL) or output spaces (for TIL and CIL, where TIL requires the test-time oracle of task identities) are often assumed to be disjoint. Specifically, $\forall i, j \in \{1, ..., T\}, i \neq j, \mathcal{Y}_i = \mathcal{Y}_j, \mathcal{X}_i \cap \mathcal{X}_j = \emptyset$ for DIL. $\forall i, j \in \{1, ..., T\}, i \neq j, \mathcal{Y}_i \cap \mathcal{Y}_j = \emptyset$ for TIL and CIL. Also, each $\mathcal{D}_t$ is learned in an

---

[1]Due to the space limit, we present a comprehensive summary of related work in Appendix A.

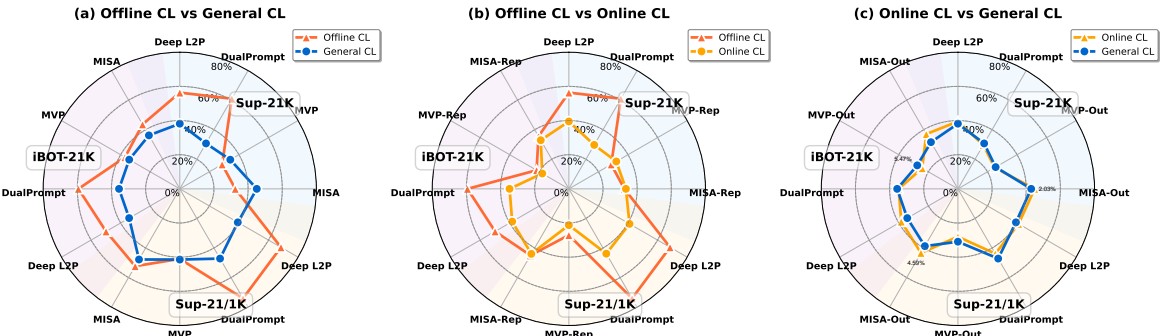

*Figure 2.* Empirical analysis of PTMs-based methods under different experimental setups. We compare (a) Offline CL vs General CL, (b) Offline CL vs Online CL, and (c) Online CL vs General CL. "-Rep", without logit masking. "-Out", without representation learning.

*offline CL* manner, i.e., the model learns all data-label pairs $(\boldsymbol{x}_t, y_t) \in \mathcal{D}_t$ over multiple epochs till convergence. In contrast, *online CL* and GCL often assumes all tasks to be learned with a one-pass online datastream, i.e., only one epoch, which poses the challenge of rapid adaptation. Meanwhile, GCL involves blurry task boundaries that the label spaces are different but may overlapped across tasks, making it difficult to balance the task-wise knowledge:

$$\forall i,j \in \{1,...,T\}, i \neq j, P(\mathcal{Y}_i \cap \mathcal{Y}_j \neq \emptyset) \geq 0, \quad (1)$$

where $P(\cdot)$ denotes the overlapping probability. GCL also includes other practical challenges such as "no test-time oracle" and "constant memory" (De Lange et al., 2021; Buzzega et al., 2020), which are not explicitly formulated for clarity. **Si-Blurry** (Moon et al., 2023) is a recent GCL setting that incorporates the above challenges. It divides classes of the overall label space $\mathcal{Y}$ into disjoint classes of $\mathcal{Y}^D$ and blurry classes of $\mathcal{Y}^B$, where $\mathcal{Y} = \mathcal{Y}^D \cup \mathcal{Y}^B$ and $\mathcal{Y}^D \cap \mathcal{Y}^B = \emptyset$. The *disjoint class ratio* $m = |\mathcal{Y}^D|/|\mathcal{Y}|$ regulates the proportion of disjoint classes. $\mathcal{Y}^D$ and $\mathcal{Y}^B$ are assigned to sequential tasks in a *non-uniform* manner, i.e., $\{\mathcal{Y}_t^D\}_{t=1...T}$ and $\{\mathcal{Y}_t^B\}_{t=1...T}$ where $\mathcal{Y}_i^D \cap \mathcal{Y}_j^D = \emptyset, P(\mathcal{Y}_i^B \cap \mathcal{Y}_j^B \neq \emptyset) \geq 0, \forall i,j \in \{1,...,T\}$ and $i \neq j$. The training samples of $\mathcal{Y}_t^D$ are all introduced when learning task $t$, while the training samples of $\mathcal{Y}_t^B$ are assigned to sequential tasks with a *blurry sample ratio* $n$. Therefore, $m$ and $n$ control the task sequence of Si-Blurry. This formulation has proven to satisfy Eq. (1) as a realization of GCL (Kang et al., 2025).

### 2.2. Empirical Analysis of PTMs-Based Methods

Although recent PTMs-based methods have made significant progress with strong supervised PTMs, their designs in both representation learning and output alignment remain sub-optimal in addressing the GCL challenges, especially under self-supervised PTMs that are more realistic yet often underfitted in their representations. Here we provide an in-depth empirical investigation of mainstream PTMs-based CL and GCL methods, with 5-task ImageNet-R as

the benchmark (Fig. 2) with details in Sec. B. We compare three groups of settings to dissect the distinct impact of online datastreams and blurry task boundaries: offline CL vs GCL, offline CL vs online CL, and online CL vs GCL. We consider three representative pretrained checkpoints: Sup-21K (supervised pretraining on ImageNet-21K), Sup-21/1K (self-supervised pretraining on ImageNet-21K and supervised finetuning on ImageNet-1K), and iBOT-21K (self-supervised pretraining on ImageNet-21K).

Overall, PTMs-based CL methods such as L2P (Wang et al., 2022b) and DualPrompt (Wang et al., 2022a) perform well in the offline setting, but their performance markedly drops once moved to GCL (Fig. 2a). These methods rely on repeatedly refining prompts, a process that becomes substantially less effective under single-pass online updates (Fig. 2b), explaining the majority of their degradation when transitioning from offline CL to GCL (Fig. 2a–c). By contrast, PTMs-based GCL methods such as MVP (Moon et al., 2023) and MISA (Kang et al., 2025) show more stable behavior across the three settings, particularly when strong supervised PTMs are used. However, their robustness diminishes notably when shifting to the more challenging self-supervised PTMs, where feature separability is weaker and adaptation under online CL and GCL constraints becomes harder (Fig. 2a).

We further dissect the designs of PTMs-based GCL methods for **representation learning** ("-Rep") and **output alignment** ("-Out"). MVP devises a contrastive loss for visual prompt tuning, but is less effective in addressing online datastreams under Sup-21/1K and iBOT-21K (MVP-Rep, Fig. 2b). While its learnable logit mask performs even better with blurry task boundaries, the baseline performance is extremely low (MVP-Out, Fig. 2c). On the other hand, MISA achieves state-of-the-art GCL performance through the initialization of prompt parameters (MISA-Rep, Fig. 2b) and non-parametric logit mask (MISA-Out, Fig. 2c). However, MISA-Rep fails to improve the representation learning of its baseline method under Sup-21/1K and iBOT-21K (compared to DualPrompt, Fig. 2b), and MISA-Out suffers

clear performance degradation with blurry task boundaries (-2.03%, -4.59%, and -5.47% on Sup-21K, Sup-21/1K, and iBOT-21K, respectively, Fig. 2c). These observations motivate us to explore more effective strategies for representation learning from online datastreams and output alignment from blurry task boundaries, as described in the following section.

## 3. Meta Post-Refinement

In this section, we present an innovative approach named **Me**ta **Po**st-Refinement (MePo) for PTMs-based GCL (Fig. 1, pseudo-code in Appendix Alg. 1). Our approach involves a meta-learning framework with subsets of pretraining data, which facilitates rapid adaptation of pretrained representations to GCL (**Meta Rep**) and initializes a meta covariance matrix for robust output alignment (**Meta Cov**).

### 3.1. MePo for Representation Learning

Due to the discrepancy between the pretraining and GCL objectives, mainstream PET techniques often struggle to capture the nuances of online datastreams. Initialization of prompt parameters (Kang et al., 2025) has been shown to be an effective strategy, but is still limited by their tuning capacity and catastrophic forgetting, resulting in sub-optimal performance especially under the more realistic self-supervised PTMs (Sec. 2.2). Inspired by meta-plasticity (Abraham, 2008; Abraham & Bear, 1996) underlying the brain networks, which retain substantial "pre-trained knowledge" positioned in a critical state of neurodynamics for rapid adaptation, we propose a MePo framework to improve the adaptability of the entire backbone parameters $\theta$ to downstream GCL tasks. Our framework constructs pseudo task sequences from pretraining data and develops a bi-level meta-learning paradigm: an inner loop simulates sequentially arriving tasks and an outer loop optimizes meta-level generalization (see Fig. 1), thereby obtaining GCL-tailored representations in a data-driven manner.

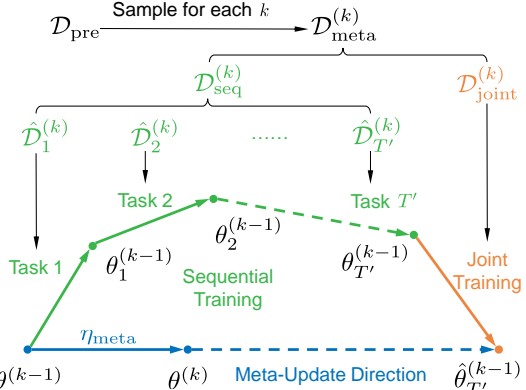

*Figure 3.* MePo representation learning.

**Pseudo Task Sequence.** The bi-level meta-learning paradigm allows for data-driven inductive bias through the specialized design of its learning objective and task sampling (Finn et al., 2017; Javed & White, 2019). In our case, the objective is to ensure rapid adaptation of pretrained representations to the online datastream in GCL. Since the true task sequence $\mathcal{D}_1, \ldots, \mathcal{D}_T$ is not available during the pretraining stage, we propose to construct pseudo task sequences $\hat{\mathcal{D}}_1, \ldots, \hat{\mathcal{D}}_{T'}$ from the pretraining dataset $\mathcal{D}_{\text{pre}}$. Specifically, in each meta-epoch $k \in \{1, \ldots, K\}$, we randomly sample a subset $\mathcal{D}_{\text{meta}}^{(k)} \in \mathcal{D}_{\text{pre}}$ consisting of classes $c \in \mathcal{C}_{\text{meta}}$ with $N_{\text{meta}}^c$ training samples per class. Then, $\mathcal{D}_{\text{meta}}^{(k)}$ is partitioned into a meta-training set $\mathcal{D}_{\text{seq}}^{(k)}$ and a meta-validation set $\mathcal{D}_{\text{joint}}^{(k)}$ according to a training-validation split ratio $\gamma$ of the class-wise training samples. The pseudo task sequences $\hat{\mathcal{D}}_1^{(k)}, \ldots, \hat{\mathcal{D}}_{T'}^{(k)}$ are constructed by randomly splitting the class set $\mathcal{C}_{\text{meta}}$ in a similar way as described in Sec. 2. Then, we formulate the bi-level optimization as $\theta^{(k)} = \arg\min_\theta \mathcal{L}(\theta, \mathcal{D}_{\text{joint}}^{(k)})$, s.t., $\theta = \text{InnerLoop}(\theta^{(k-1)}, \mathcal{D}_{\text{seq}}^{(k)})$.

**Inner Loop: Sequential Training.** Given the pseudo task sequence sampled at each meta-epoch $k \in \{1, \ldots, K\}$, we update both $\theta$ and $\psi$ by learning sequentially arriving tasks $t \in \{1, \ldots, T'\}$ with the task-specific loss:

$$\mathcal{L}_t(\theta, \psi) = \mathbb{E}_{(\boldsymbol{x}, y) \sim \hat{\mathcal{D}}_t^{(k)}}[\mathcal{L}_{\text{CE}}(h_\psi(f_\theta(\boldsymbol{x})), y)], \quad (2)$$

where $\mathcal{L}_{\text{CE}}$ is the cross-entropy loss for classification tasks. We further denote $\theta_{t-1}^{(k-1)}$ as the backbone parameters updated from the meta-epoch $k - 1$ after learning task $t - 1$, where $\theta^{(0)}$ represents the original pretrained backbone parameters (the task identity is omitted if the pseudo task sequence has not yet been introduced). Similarly, we denote $\psi_{t-1}$ as the output layer parameters after learning task $t - 1$. With learning rates $\eta_\theta$ and $\eta_\psi$, the entire model is sequentially optimized as

$$\theta_t^{(k-1)} = \theta_{t-1}^{(k-1)} - \eta_\theta \nabla_\theta \mathcal{L}_t(\theta_{t-1}^{(k-1)}, \psi_{t-1}), \quad (3)$$

$$\psi_t = \psi_{t-1} - \eta_\psi \nabla_\psi \mathcal{L}_t(\theta_{t-1}^{(k-1)}, \psi_{t-1}). \quad (4)$$

**Outer Loop: Joint Training.** After performing the inner loop, we refine the backbone parameters $\theta_{T'}^{(k-1)}$ by joint training of all tasks with the held-out meta-validation set $\mathcal{D}_{\text{joint}}^{(k)}$, which encourages the pretrained representations to overcome potential bias caused by sequential training:

$$\hat{\theta}_{T'}^{(k-1)} = \theta_{T'}^{(k-1)} - \eta_\theta \nabla_\theta \mathbb{E}_{(\boldsymbol{x}, y) \sim \mathcal{D}_{\text{joint}}^{(k)}}[\mathcal{L}_{\text{CE}}(h_{\psi_{T'}}(f_{\theta_{T'}^{(k-1)}}(\boldsymbol{x})), y)]. \quad (5)$$

With $\hat{\theta}_{T'}^{(k-1)}$ obtained from the bi-level learning paradigm, we follow the previous work (Nichol & Schulman, 2018) to accumulate parameter updates through a first-order approximation:

$$\theta^{(k)} = \theta^{(k-1)} + \eta_{\text{meta}}(\hat{\theta}_{T'}^{(k-1)} - \theta^{(k-1)}), \quad (6)$$

where $\eta_{\text{meta}} \in [0, 1]$ denotes the meta-learning rate. This update encourages the pretrained backbone to evolve towards representations that are appropriate for learning a potentially new task sequence in GCL (see Fig. 3). The entire parameter updates persist for $K$ meta-epochs, culminating in the refined backbone parameters $\theta^*$ for output alignment, as described below.

**Mechanism of Meta Rep.** Meta-learning methods such as MAML (Finn et al., 2017) aim to learn an initialization that enables rapid adaptation. Reptile (Nichol et al., 2018) further demonstrates that this can be achieved through a first-order approximation to the second-order meta-gradient. Following this theoretical principle, MePo Rep constructs pseudo sequential tasks from pretraining data so that the inner loop simulates CL-style sequential updates. The outer loop then meta-refines the backbone to remain stable after these updates, yielding a CL-tailored initialization that is resilient to sequential drift yet retains plasticity that standard finetuning cannot provide (see detailed proof in Sec. D).

### 3.2. MePo for Output Alignment

With $\theta^*$, we strive to further rectify the potential bias of the output layer. Recent PTMs-based GCL methods (Moon et al., 2023; Kang et al., 2025) often involve logit masking of classes observed in each batch, yet limited by the over simplified representation modeling (i.e., the output layer amounts to preserving class-wise prototypes) and severely imbalanced classes in GCL. Advanced PTMs-based CL methods (McDonnell et al., 2024; Zhang et al., 2023; Wang et al., 2023) have identified that the second-order statistics (i.e., the feature covariance) are critical for preserving the geometry of representation space to obtain well-balanced predictions, but are difficult to estimate all at once in GCL. Inspired by the reconstructive memory theory (Lei et al., 2022; 2024; Richards & Frankland, 2017) in neuroscience, where the neural representations of incoming memories are continually encoded into and reconstructed from a previously established representation space, we propose to initialize a meta covariance matrix from pretraining data, serving as a reference geometry for robust output alignment.

**Meta Covariance Matrix.** To approximate the second-order statistics of pretrained representations, we randomly sample a reference group of class-specific subsets $\mathcal{D}_{\text{ref}} = \{\mathcal{D}_{\text{ref}}^c\}_{c \in \mathcal{C}_{\text{ref}}} \in \mathcal{D}_{\text{pre}}$ consisting of classes $c \in \mathcal{C}_{\text{ref}}$ with $N_{\text{ref}}^c$ training samples per class. We then obtain the class-wise feature mean:

$$\boldsymbol{\mu}_c = \frac{1}{N_c} \sum_{i=1}^{N_c} f_{\theta^*}(\boldsymbol{x}_i), \quad (\boldsymbol{x}_i, c) \in \mathcal{D}_{\text{ref}}^c. \qquad (7)$$

Next, we obtain the covariance matrix initialized with $\mathcal{D}_{\text{ref}}$ as a reference geometry of pretrained representation space,

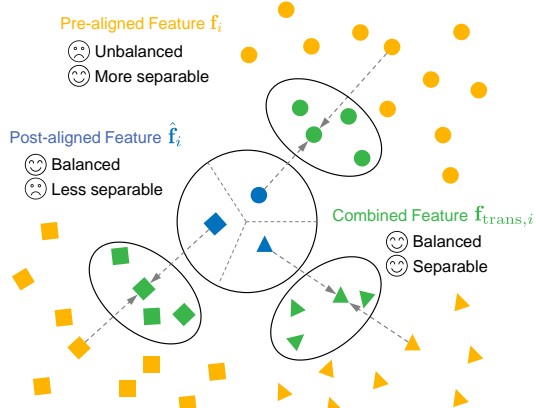

*Figure 4.* MePo feature alignment.

which is subsequently used in GCL to perform output alignment:

$$\boldsymbol{\Sigma}_{\text{pre}} = \frac{1}{|\mathcal{C}_{\text{ref}}| - 1} \sum_{c=1}^{|\mathcal{C}_{\text{ref}}|} (\boldsymbol{\mu}_c - \bar{\boldsymbol{\mu}})(\boldsymbol{\mu}_c - \bar{\boldsymbol{\mu}})^\top, \qquad (8)$$

where $\bar{\boldsymbol{\mu}} = \frac{1}{|\mathcal{C}_{\text{ref}}|} \sum_{c=1}^{|\mathcal{C}_{\text{ref}}|} \boldsymbol{\mu}_c$ denotes the global feature mean.

**Feature Alignment.** During the GCL phase, training samples are introduced in small batches of imbalanced classes. To balance their contributions, we first calculate batch-wise feature covariance from the feature vector $\mathbf{f}_i = f_{\theta^* + \Delta\theta}(\boldsymbol{x}_i)$ of each training sample $(\boldsymbol{x}_i, y_i) \in \mathcal{D}_t$, where $\Delta\theta$ denotes the tunable parameters for representation learning in GCL (usually implemented via PET techniques). Given a batch of training samples alongside features $\mathcal{B} = \{\mathbf{f}_i\}_{i=1}^{|\mathcal{B}|}$, we estimate the batch-wise feature mean and covariance:

$$\bar{\mathbf{f}} = \frac{1}{|\mathcal{B}|} \sum_{i=1}^{|\mathcal{B}|} \mathbf{f}_i, \quad \boldsymbol{\Sigma}_{\text{cur}} = \frac{1}{|\mathcal{B}| - 1} \sum_{i=1}^{|\mathcal{B}|} (\mathbf{f}_i - \bar{\mathbf{f}})(\mathbf{f}_i - \bar{\mathbf{f}})^\top. \tag{9}$$

To rectify the potential bias of imbalanced classes in GCL, we propose to align batch-wise feature distributions (i.e., $\boldsymbol{\Sigma}_{\text{cur}}$) to the reference geometry of pretrained representation space (i.e., $\boldsymbol{\Sigma}_{\text{pre}}$) for subsequent use in prediction. Here we align $\boldsymbol{\Sigma}_{\text{cur}}$ and $\boldsymbol{\Sigma}_{\text{pre}}$ via the Cholesky decomposition (Benoit, 1924; Watkins, 2004; Press, 1992), an efficient and numerically stable strategy to decompose a positive definite matrix (e.g., the covariance matrix) into the product of a lower triangular matrix and its transpose. We calculate the lower triangular matrices of current feature statistics $\boldsymbol{L}_{\text{cur}}$ by decomposing $\boldsymbol{\Sigma}_{\text{cur}} = \boldsymbol{L}_{\text{cur}} \boldsymbol{L}_{\text{cur}}^\top$ and of pretrained feature statistics $\boldsymbol{L}_{\text{pre}}$ by decomposing $\boldsymbol{\Sigma}_{\text{pre}} = \boldsymbol{L}_{\text{pre}} \boldsymbol{L}_{\text{pre}}^\top$. We then align each feature vector $\mathbf{f}_i$ to the pretrained representation space:

$$\hat{\mathbf{f}}_i = \mathbf{f}_i \boldsymbol{A}, \quad \boldsymbol{A} = \boldsymbol{L}_{\text{cur}}^{-1} \boldsymbol{L}_{\text{pre}}, \qquad (10)$$

which ensures $\mathbb{E}_{\mathbf{f}_i \in \mathcal{B}}[\hat{\mathbf{f}}_i \hat{\mathbf{f}}_i^\top] = \boldsymbol{A}^\top \boldsymbol{\Sigma}_{\text{cur}} \boldsymbol{A} = \boldsymbol{\Sigma}_{\text{pre}}$.

*Table 1.* Overall performance of different methods in GCL. All results are averaged over five runs ($\pm$ standard deviation) with different task sequences.

| PTM | Method | CIFAR-100 | | ImageNet-R | | CUB-200 | |
|---|---|---|---|---|---|---|---|
| | | $A_{\mathrm{AUC}}(\uparrow)$ | $A_{\mathrm{Last}}(\uparrow)$ | $A_{\mathrm{AUC}}(\uparrow)$ | $A_{\mathrm{Last}}(\uparrow)$ | $A_{\mathrm{AUC}}(\uparrow)$ | $A_{\mathrm{Last}}(\uparrow)$ |
| Sup-21K | Seq FT | $19.71_{\pm3.39}$ | $10.42_{\pm4.92}$ | $7.51_{\pm3.94}$ | $2.29_{\pm0.85}$ | $3.47_{\pm0.41}$ | $1.49_{\pm0.42}$ |
| | Linear Probe | $49.69_{\pm6.09}$ | $23.07_{\pm7.33}$ | $29.24_{\pm1.26}$ | $16.87_{\pm3.14}$ | $28.96_{\pm2.46}$ | $17.33_{\pm3.08}$ |
| | Seq FT (SL) | $64.90_{\pm7.18}$ | $62.06_{\pm1.89}$ | $47.20_{\pm1.47}$ | $39.60_{\pm2.43}$ | $56.16_{\pm4.32}$ | $56.50_{\pm3.08}$ |
| | CODA-P | $78.81_{\pm3.38}$ | $80.30_{\pm1.58}$ | $50.11_{\pm2.14}$ | $46.17_{\pm2.00}$ | $64.96_{\pm3.30}$ | $59.28_{\pm3.14}$ |
| | Deep L2P | $78.12_{\pm0.61}$ | $77.73_{\pm1.09}$ | $42.39_{\pm0.23}$ | $38.16_{\pm1.37}$ | $60.95_{\pm1.22}$ | $56.31_{\pm2.53}$ |
| | w/ MePo (Ours) | $\mathbf{83.63}_{\pm0.61}$ | $\mathbf{83.98}_{\pm0.29}$ | $\mathbf{58.71}_{\pm1.28}$ | $\mathbf{55.13}_{\pm1.16}$ | $\mathbf{64.92}_{\pm1.47}$ | $\mathbf{63.30}_{\pm1.52}$ |
| | DualPrompt | $66.36_{\pm4.42}$ | $58.09_{\pm4.40}$ | $38.63_{\pm2.19}$ | $30.71_{\pm0.82}$ | $55.73_{\pm2.77}$ | $47.08_{\pm4.94}$ |
| | w/ MePo (Ours) | $\mathbf{71.37}_{\pm4.07}$ | $\mathbf{66.48}_{\pm2.82}$ | $\mathbf{44.65}_{\pm2.09}$ | $\mathbf{36.76}_{\pm1.21}$ | $\mathbf{58.36}_{\pm2.59}$ | $\mathbf{52.16}_{\pm3.74}$ |
| | MVP | $68.13_{\pm4.34}$ | $60.56_{\pm2.57}$ | $41.50_{\pm1.15}$ | $34.14_{\pm3.95}$ | $56.78_{\pm2.88}$ | $50.25_{\pm3.53}$ |
| | w/ MePo (Ours) | $\mathbf{72.18}_{\pm4.50}$ | $\mathbf{68.45}_{\pm1.59}$ | $\mathbf{46.35}_{\pm1.31}$ | $\mathbf{38.21}_{\pm3.66}$ | $\mathbf{58.73}_{\pm3.31}$ | $\mathbf{52.22}_{\pm2.80}$ |
| | MISA | $80.35_{\pm2.39}$ | $80.75_{\pm1.24}$ | $51.52_{\pm2.09}$ | $45.08_{\pm1.43}$ | $65.40_{\pm3.01}$ | $60.20_{\pm1.82}$ |
| | w/ MePo (Ours) | $\mathbf{82.30}_{\pm2.83}$ | $\mathbf{83.99}_{\pm1.35}$ | $\mathbf{54.86}_{\pm2.20}$ | $\mathbf{49.18}_{\pm1.38}$ | $\mathbf{68.13}_{\pm3.17}$ | $\mathbf{64.75}_{\pm1.00}$ |
| Sup-21/1K | Deep L2P | $69.15_{\pm1.66}$ | $68.57_{\pm1.38}$ | $42.74_{\pm0.83}$ | $39.22_{\pm2.14}$ | $39.20_{\pm1.69}$ | $46.76_{\pm1.87}$ |
| | w/ MePo (Ours) | $\mathbf{78.75}_{\pm1.18}$ | $\mathbf{77.52}_{\pm1.03}$ | $\mathbf{62.71}_{\pm1.09}$ | $\mathbf{58.91}_{\pm0.08}$ | $\mathbf{48.36}_{\pm1.88}$ | $\mathbf{50.88}_{\pm2.85}$ |
| | DualPrompt | $64.84_{\pm2.62}$ | $67.22_{\pm8.54}$ | $49.52_{\pm2.92}$ | $47.14_{\pm3.39}$ | $43.96_{\pm2.00}$ | $41.20_{\pm7.61}$ |
| | w/ MePo (Ours) | $\mathbf{67.18}_{\pm4.48}$ | $\mathbf{57.95}_{\pm3.69}$ | $\mathbf{54.75}_{\pm1.66}$ | $\mathbf{44.75}_{\pm0.74}$ | $\mathbf{47.06}_{\pm3.19}$ | $\mathbf{38.24}_{\pm9.29}$ |
| | MVP | $65.26_{\pm3.87}$ | $53.66_{\pm5.61}$ | $51.26_{\pm1.47}$ | $41.41_{\pm4.81}$ | $45.12_{\pm3.08}$ | $37.95_{\pm9.32}$ |
| | w/ MePo (Ours) | $\mathbf{70.25}_{\pm4.23}$ | $\mathbf{62.05}_{\pm2.39}$ | $\mathbf{61.28}_{\pm1.21}$ | $\mathbf{50.82}_{\pm3.70}$ | $\mathbf{49.72}_{\pm3.53}$ | $\mathbf{42.81}_{\pm6.74}$ |
| | MISA | $62.91_{\pm7.96}$ | $67.99_{\pm7.41}$ | $50.87_{\pm1.69}$ | $47.75_{\pm2.87}$ | $42.76_{\pm2.33}$ | $44.05_{\pm1.94}$ |
| | w/ MePo (Ours) | $\mathbf{78.01}_{\pm3.09}$ | $\mathbf{76.73}_{\pm1.06}$ | $\mathbf{64.23}_{\pm1.30}$ | $\mathbf{58.20}_{\pm0.51}$ | $\mathbf{55.31}_{\pm4.52}$ | $\mathbf{56.58}_{\pm2.33}$ |
| iBOT-21K | Deep L2P | $64.48_{\pm1.23}$ | $66.71_{\pm1.27}$ | $33.68_{\pm2.78}$ | $36.24_{\pm1.83}$ | $16.22_{\pm0.85}$ | $27.14_{\pm0.75}$ |
| | w/ MePo (Ours) | $\mathbf{75.83}_{\pm1.23}$ | $\mathbf{76.40}_{\pm0.94}$ | $\mathbf{55.30}_{\pm0.50}$ | $\mathbf{52.38}_{\pm1.87}$ | $\mathbf{40.90}_{\pm1.44}$ | $\mathbf{46.50}_{\pm2.90}$ |
| | DualPrompt | $63.09_{\pm2.36}$ | $61.20_{\pm8.76}$ | $41.33_{\pm2.11}$ | $35.58_{\pm3.24}$ | $24.56_{\pm2.25}$ | $21.32_{\pm6.38}$ |
| | w/ MePo (Ours) | $\mathbf{65.76}_{\pm3.56}$ | $\mathbf{59.21}_{\pm3.18}$ | $\mathbf{48.06}_{\pm2.20}$ | $\mathbf{37.69}_{\pm2.10}$ | $\mathbf{38.19}_{\pm3.74}$ | $\mathbf{31.03}_{\pm11.55}$ |
| | MVP | $64.01_{\pm3.27}$ | $50.00_{\pm11.45}$ | $43.89_{\pm1.88}$ | $34.19_{\pm4.56}$ | $29.59_{\pm3.28}$ | $27.85_{\pm8.89}$ |
| | w/ MePo (Ours) | $\mathbf{66.88}_{\pm4.86}$ | $\mathbf{57.19}_{\pm2.63}$ | $\mathbf{53.75}_{\pm1.38}$ | $\mathbf{42.55}_{\pm3.08}$ | $\mathbf{40.99}_{\pm3.45}$ | $\mathbf{34.66}_{\pm8.40}$ |
| | MISA | $65.30_{\pm2.28}$ | $67.43_{\pm6.75}$ | $40.94_{\pm1.22}$ | $36.16_{\pm1.58}$ | $18.62_{\pm3.36}$ | $23.66_{\pm2.21}$ |
| | w/ MePo (Ours) | $\mathbf{75.80}_{\pm3.77}$ | $\mathbf{76.02}_{\pm1.18}$ | $\mathbf{57.00}_{\pm2.52}$ | $\mathbf{49.86}_{\pm1.22}$ | $\mathbf{49.33}_{\pm3.59}$ | $\mathbf{45.68}_{\pm2.59}$ |

The pre-aligned feature $\mathbf{f}_i$ and the post-aligned feature $\hat{\mathbf{f}}_i$ exhibit distinct properties (see Fig. 4): $\mathbf{f}_i$ collected from the finetuned representation space of $f_{\theta^*+\Delta\theta}(\cdot)$ tends to be more separable yet imbalanced, while $\hat{\mathbf{f}}_i$ aligned to the pretrained representation space of $f_{\theta^*}(\cdot)$ tend to be more balanced yet crowded. We take advantages of both via a weighted combination:

$$\mathbf{f}_{\mathrm{trans},i} = \alpha\hat{\mathbf{f}}_i + (1-\alpha)\mathbf{f}_i, \qquad (11)$$

where $\alpha \in [0,1]$ is a hyperparameter that controls the balance of stability and plasticity.

Finally, we employ the combined feature $\mathbf{f}_{\mathrm{trans},i}$ to update the tunable backbone parameters $\Delta\theta$ and the output layer parameters $\psi$ during the GCL phase:

$$\mathcal{L}_{\mathrm{CE}}(h_\psi(\mathbf{f}_{\mathrm{trans},i}), y_i) = -\sum_{c\in\mathcal{Y}_t} y_i^{(c)} \log p_i^{(c)},$$
$$p_i^{(c)} = \frac{\exp(h_\psi(\mathbf{f}_{\mathrm{trans},i})^{(c)})}{\sum_{k\in\mathcal{Y}_t} \exp(h_\psi(\mathbf{f}_{\mathrm{trans},i})^{(k)})}, \qquad (12)$$

where the superscript $(c)$ denotes the vector component corresponding to class $c$.

**Mechanism of Meta Cov.** Meta Cov addresses the challenge that feature covariance in PTMs-based CL drifts severely under small, noisy, and imbalanced online batches, leading to distorted representation geometry and increased task interference. To stabilize this process, Meta Cov introduces a meta covariance matrix $\Sigma_{pre}$ computed from large-scale, balanced pretraining data, serving as a reliable reference geometry. By aligning $\Sigma_{cur}$ toward $\Sigma_{pre}$ through a Cholesky transformation, Meta Cov constrains feature updates to a stable manifold, preventing collapse or expansion and improving the overall stability–plasticity balance.

## 4. Experiments

In this section, we will first describe the experimental setups (further detailed in Appendix Sec. B) of GCL with Si-Blurry, including datasets, baseline methods, evaluation metrics and training details, and then present the experimental results with an in-depth analysis.

**Overall Performance.** We first evaluate the overall performance in Table 1. MISA is the state-of-the-art GCL method

*Table 2.* Ablation study of representation (Meta Rep) and covariance (Meta Cov) in MePo. All results are averaged over five runs ($\pm$ standard deviation) with different task sequences.

| PTM | Meta Rep | Meta Cov | ImageNet-R (MVP) | | ImageNet-R (MISA) | | CUB-200 (MVP) | | CUB-200 (MISA) | |
|---|---|---|---|---|---|---|---|---|---|---|
| | | | $A_{\mathrm{AUC}}(\uparrow)$ | $A_{\mathrm{Last}}(\uparrow)$ | $A_{\mathrm{AUC}}(\uparrow)$ | $A_{\mathrm{Last}}(\uparrow)$ | $A_{\mathrm{AUC}}(\uparrow)$ | $A_{\mathrm{Last}}(\uparrow)$ | $A_{\mathrm{AUC}}(\uparrow)$ | $A_{\mathrm{Last}}(\uparrow)$ |
| Sup-21K | ✗ | ✗ | $41.50_{\pm1.15}$ | $34.14_{\pm3.95}$ | $51.52_{\pm2.09}$ | $45.08_{\pm1.43}$ | $56.78_{\pm2.88}$ | $50.25_{\pm3.53}$ | $65.40_{\pm3.01}$ | $60.20_{\pm1.82}$ |
| | ✓ | ✗ | $46.32_{\pm1.29}$ | $38.06_{\pm3.77}$ | $52.35_{\pm2.09}$ | $45.81_{\pm1.08}$ | $58.67_{\pm2.80}$ | $51.65_{\pm3.23}$ | $64.83_{\pm2.82}$ | $59.57_{\pm1.73}$ |
| | ✗ | ✓ | $40.51_{\pm1.20}$ | $33.99_{\pm4.11}$ | $53.59_{\pm2.26}$ | $47.90_{\pm1.65}$ | $55.82_{\pm3.64}$ | $51.49_{\pm2.72}$ | $68.03_{\pm3.05}$ | $\mathbf{65.30}_{\pm1.82}$ |
| | ✓ | ✓ | $\mathbf{46.35}_{\pm1.31}$ | $\mathbf{38.21}_{\pm3.66}$ | $\mathbf{54.86}_{\pm2.20}$ | $\mathbf{49.18}_{\pm1.38}$ | $\mathbf{58.73}_{\pm3.31}$ | $\mathbf{52.22}_{\pm2.80}$ | $\mathbf{68.13}_{\pm3.17}$ | $64.75_{\pm1.00}$ |
| Sup-21/1K | ✗ | ✗ | $51.26_{\pm1.47}$ | $41.41_{\pm4.81}$ | $50.87\pm1.69$ | $47.75\pm2.87$ | $45.12_{\pm3.08}$ | $37.95_{\pm9.32}$ | $42.76\pm2.33$ | $44.05\pm1.94$ |
| | ✓ | ✗ | $57.50_{\pm1.18}$ | $46.75_{\pm4.85}$ | $56.71\pm1.08$ | $50.29\pm2.04$ | $47.26_{\pm3.38}$ | $39.65_{\pm8.09}$ | $44.68\pm2.46$ | $43.88\pm2.72$ |
| | ✗ | ✓ | $55.83_{\pm1.62}$ | $45.83_{\pm5.07}$ | $57.66\pm0.96$ | $52.30\pm0.58$ | $47.69_{\pm3.05}$ | $40.51_{\pm8.67}$ | $49.60\pm3.31$ | $47.21\pm1.72$ |
| | ✓ | ✓ | $\mathbf{61.28}_{\pm1.21}$ | $\mathbf{50.82}_{\pm3.70}$ | $\mathbf{64.23}\pm1.30$ | $\mathbf{58.20}\pm0.51$ | $\mathbf{49.72}_{\pm3.53}$ | $\mathbf{42.81}_{\pm6.74}$ | $\mathbf{55.31}\pm4.52$ | $\mathbf{56.58}\pm2.33$ |
| iBOT-21K | ✗ | ✗ | $43.89_{\pm1.88}$ | $34.19_{\pm4.56}$ | $40.94_{\pm1.22}$ | $36.16_{\pm1.58}$ | $29.59_{\pm3.28}$ | $27.85_{\pm8.89}$ | $18.62_{\pm3.36}$ | $23.66_{\pm2.21}$ |
| | ✓ | ✗ | $52.67_{\pm1.41}$ | $41.91_{\pm3.95}$ | $50.21_{\pm1.93}$ | $43.52_{\pm1.39}$ | $40.61_{\pm3.41}$ | $34.17_{\pm9.09}$ | $39.44_{\pm2.93}$ | $40.38_{\pm1.79}$ |
| | ✗ | ✓ | $47.44_{\pm1.76}$ | $37.02_{\pm5.00}$ | $44.24_{\pm1.90}$ | $38.08_{\pm1.15}$ | $31.75_{\pm3.43}$ | $30.35_{\pm9.12}$ | $20.65_{\pm3.22}$ | $22.92_{\pm0.75}$ |
| | ✓ | ✓ | $\mathbf{53.75}_{\pm1.38}$ | $\mathbf{42.55}_{\pm3.08}$ | $\mathbf{57.00}_{\pm2.52}$ | $\mathbf{49.86}_{\pm1.22}$ | $\mathbf{40.99}_{\pm3.45}$ | $\mathbf{34.66}_{\pm8.40}$ | $\mathbf{49.33}_{\pm3.59}$ | $\mathbf{45.68}_{\pm2.59}$ |

that outperforms other PTMs-based CL and GCL methods under strong supervised PTMs (Sup-21K) and general datasets (CIFAR-100 and ImageNet-R). However, the performance of all baselines tend to decay severely under weakly supervised and self-supervised PTMs (Sup-21/1K and iBOT-21K) and fine-grained dataset (CUB-200), both of which strengthen the challenges of representation learning and output alignment in GCL. Interestingly, the re-implemented Deep L2P achieves competing or even better performance than PTMs-based GCL baselines in many cases, suggesting limited progress of the current GCL research.

In comparison, our proposed MePo serves as a plug-in strategy that substantially enhances the performance of PTMs-based CL and GCL methods in Si-Blurry (Table 1), traditional online CL, offline CL and domain CL settings (Appendix Table 7). The performance gains tend to be more significant from supervised to self-supervised PTMs and from general to fine-grained datasets (Table 1), as well as OOD datasets NCH for chest X-ray and GTSRB for traffic sign (Appendix Table 6), suggesting the adaptive effectiveness of MePo in overcoming GCL challenges. For example, the $A_{\mathrm{AUC}}/A_{\mathrm{Last}}$ improvements over MISA are 15.10%/8.74%, 13.36%/10.45%, and 12.56%/12.53% on CIFAR-100, ImageNet-R, and CUB-200 under Sup-21/1K, demonstrating clearly the new state-of-the-art. MePo remains consistently effective across different downstream task lengths $T$ (Appendix Table 10), where using more pseudo tasks in the meta-refinement phase is more advantageous if the downstream task sequence is longer.

**Computational cost.** The computation cost of MePo consists of two components: a one-time meta-refinement phase and the subsequent downstream GCL phase. Notably, our meta-refinement can be seen as a prolonged pretraining phase with one-time cost, which is method-agnostic and reusable for GCL. Once the backbone is refined from pseudo tasks of the pretraining data, it can be directly reused by any downstream GCL method and dataset. The data bud-

gets of meta-refinement is only accounting for 0.15% of ViT-B/16 pretraining (Appendix Table 11 and 12). In the downstream GCL phase, MePo only preserves an additional covariance matrix of negligible storage overhead (0.67% of the ViT-B/16 backbone) and does not introduce additional computational overhead during the GCL stage (Table 3), positioning it as an efficient choice.

*Table 3.* Comparison of resource overheads: Batch time on ImageNet-R under Sup-21K.

| Method | +Param. | +Ratio | Time | Accuracy |
|---|---|---|---|---|
| MVP | 639k | 0.74% | 5.34s | 41.50 |
| w/ MePo | 1215k | 1.41% | 5.34s | 46.35 |
| MISA | 637k | 0.74% | 4.84s | 51.52 |
| w/ MePo | 1213k | 1.41% | 4.84s | 54.86 |

**Ablation Study.** We present an extensive ablation study with two comparably challenging datasets (ImageNet-R and CUB-200) under the three pretrained checkpoints, using MVP and MISA as the plug-in baselines. Overall, MePo for both representation learning (Meta Rep) and output alignment (Meta Cov) contributes to its strong performance (Table 2), validating the effectiveness of our designs. Interestingly, there exist some cases (e.g., MISA on CUB-200 under Sup-21/1K and iBOT-21K) where using either Meta Rep or Meta Cov alone is not necessarily effective, while only using both simultaneously can obtain considerable enhancements. These results demonstrate the complementary effects of Meta Rep and Meta Cov to overcome GCL challenges.

We further evaluate the impact of $\alpha$ in Eq. (11), i.e., the combination weight of pre-aligned and post-aligned features. As shown in Fig. 5, $\alpha$ is relatively insensitive and delivers strong improvements over a wide range of hyperparameter values (0.3-0.7). $\alpha = 0$ is equivalent to Meta Rep only, resulting in sub-optimal performance due to the balanced but less separable classes (Figs. 4 and 6). $\alpha = 1$ aligns all features to the pretrained representation space, failing to accommodate new distributions during the GCL

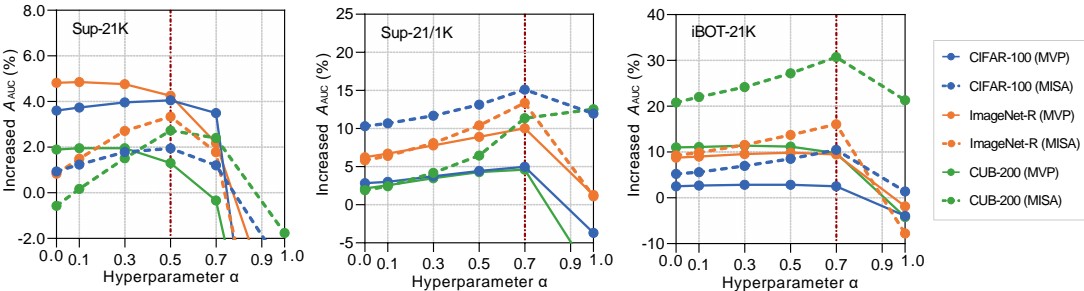

*Figure 5.* Empirical evaluation of the combination weight $\alpha$ in MePo. Here we employ $A_{\mathrm{AUC}}(\uparrow)$ as the evaluation metric. The complete quantification results are included in Appendix Tables 4 and 5.

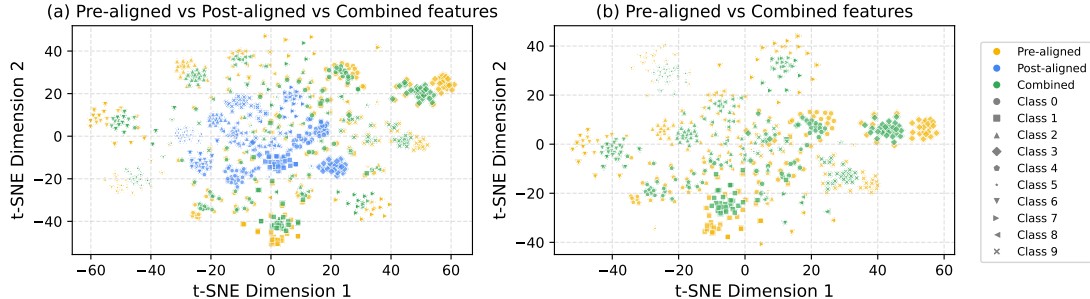

*Figure 6.* Visualization of pre-aligned, post-aligned, and combined features with t-SNE (Van der Maaten & Hinton, 2008). Here we take the setup of MISA w/ MePo, ImageNet-R, and Sup-21/1K as an example. Best viewed in color.

phase. In comparison, a moderate value strikes an appropriate balance of pretrained and finetuned representations: $\alpha = 0.5$ for Sup-21K and $\alpha = 0.7$ for Sup-21/1K and iBOT-21K, suggesting that self-supervised representations require greater stability to overcome recency bias in prediction.

**Detailed Analysis.** Here we visualize the pre-aligned, post-aligned, and combined features with t-SNE (Van der Maaten & Hinton, 2008) (Fig. 6a). The post-aligned features mapping to the pretrained representation space exhibit a "meta" distribution at the center of all features, with identical distances to the pre-aligned features of each class. The combined features generally locate between the pre-aligned and post-aligned features as the design of Eq. (11), and tend to be more separable than both. We further perform t-SNE of only pre-aligned and combined features (Fig. 6b). Again, the transformed features of each class are clearly more separable than the pre-aligned features, consistent with the significant improvements observed in Table 2 (i.e., Meta Rep with or without Meta Cov).

Next, we provide visualization results to explicitly demonstrate the effectiveness of Meta Rep and Meta Cov. We first visualize the distribution of activated class-wise representations. As shown in Fig 7, Appendix Fig. 8, and Fig. 9, the use of Meta Rep results in much sparser activation in GCL, alleviating the mutual interference of different classes in representation space during CL (Javed & White, 2019; Michieli & Zanuttigh, 2021; Pourcel et al., 2022; Shi et al., 2022). Interestingly, a previous work called OML (Javed & White,

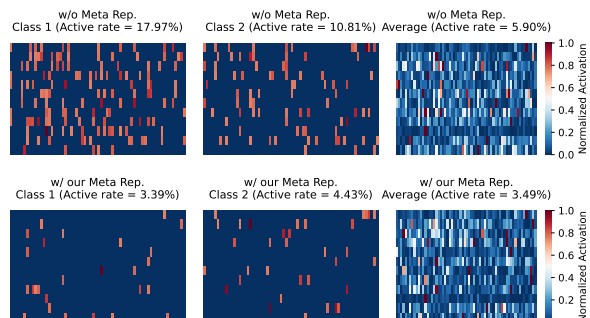

*Figure 7.* Visualization of class-wise prototypes on ImageNet-R under Sup-21K.

2019) has also attempted meta-learning representations for CL via updating the output layer and backbone parameters separately, obtaining sparser representations than naive pretraining. In comparison, Meta Rep updates all parameters within the inner loop, enabling more adequate adaptation. We empirically validate that OML is significantly inferior to ours in GCL (e.g., the $A_{\mathrm{AUC}}$ improvements over MISA are 0.58% and 6.07% with OML and Meta Rep on ImageNet-R under Sup-21/1K).

## 5. Conclusion and Discussion

In this work, we investigate GCL with Si-Blurry as a typical realization. We reveal that the two practical challenges, namely online datastreams and blurry task boundaries, severely undermine the effectiveness of advanced

PTMs-based CL and GCL methods by degrading representation learning and output alignment, respectively. To address these challenges, we propose an innovative approach that refines pretrained representations through a post-refinement process to enable rapid adaptation, and initializes a meta covariance matrix to align second-order statistics within the representation space. Our approach achieves state-of-the-art performance across an range of benchmark datasets and pretrained checkpoints. We contend that GCL scenarios mirror the highly complex and dynamic nature of real-world environments, and the effective use of post-refinement offers a promising solution. These explorations are expected to further enhance AI adaptability, such as enabling robust online interaction with the real physical world.

## Acknowledgment.

This work is supported by the Beijing Major Science and Technology Project (No. Z251100008425003), the STI2030-Major Projects (No. 2022ZD0204900), the NSFC Projects (Nos. 62406160, 92370124, U25B6003, 62350080, 62595773), the Fundamental and Interdisciplinary Disciplines Breakthrough Plan of the Ministry of Education of China (No. JYB2025XDXM101), Beijing Natural Science Foundation (No. L247011), the Shandong Provincial Natural Science Foundation (No. ZR2022ZD01), and the High Performance Computing Center, Tsinghua University.

## Impact Statement

This paper presents work whose goal is to advance the field of Machine Learning. There are many potential societal consequences of our work, none which we feel must be specifically highlighted here.

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

# A. Related Work

**Continual Learning (CL)** aims to overcome catastrophic forgetting when learning sequentially arriving tasks with distinct data distributions (Wang et al., 2024; Van de Ven & Tolias, 2019). Conventional CL settings often assume offline learning of each task with disjoint task boundaries, such as task-incremental learning (TIL), class-incremental learning (CIL), and domain-incremental learning (DIL) (Van de Ven & Tolias, 2019). Representative methods focus on CL from scratch, such as regularization-based, replay-based, and architecture-based methods. Recent advances in CL have involved PTMs to obtain better performance (Zhang et al., 2023). Since CL tends to progressively overwrite the pretrained knowledge, these methods often keep the pretrained backbone frozen and exploits PET techniques to instruct representation learning (Wang et al., 2022b;a; Wu et al., 2025). They also replay representations of old tasks to rectify potential bias of the output layer (Wang et al., 2023; McDonnell et al., 2024). However, the efficacy of PET techniques relies heavily on offline task learning with adequate training samples, and the representation replay requires disjoint task boundaries to approximate old task distributions, which severely limits their applicability.

**General Continual Learning (GCL)** is introduced to capture the practical challenges for applying CL in real-world scenarios (De Lange et al., 2021; Buzzega et al., 2020), such as "online learning", "blurry task boundaries", "no test-time oracle", "constant memory", etc. These challenges have been partially involved in many existing CL settings, such as "no test-time oracle" in CIL and "online learning" in online CL, while "constant memory" is a desirable requirement for all CL methods. Si-Blurry (Moon et al., 2023) is one of the latest GCL settings that incorporate all aforementioned challenges, where the training samples of each task are randomly sampled from distributions that may involve old and new classes. Many efforts have been made in adapting PTMs-based CL methods to this GCL scenario. For example, MVP (Moon et al., 2023) devises a contrastive loss for visual prompt tuning and adopts learnable logit masking to rectify the output layer. MISA (Kang et al., 2025) employs pretraining data to initialize the prompt parameters and simplifies the logit masking into a non-parametric implementation. Despite the promise, these methods are limited by the capacity of PET techniques for representation learning and the overly simplistic modeling of representation space for output alignment, leading to sub-optimal GCL performance.

# B. Experimental Setup

**Benchmarks.** We employ three representative datasets, CIFAR-100 (Krizhevsky et al., 2009) (general dataset, 100-class small-scale images), ImageNet-R (Hendrycks et al., 2021) (general dataset, 200-class large-scale images), and CUB-200 (Wah et al., 2011) (fine-grained dataset, 200-class large-scale images), to construct the evaluation benchmarks. We follow the official implementation of Si-Blurry (Moon et al., 2023; Kang et al., 2025), with the disjoint class ratio $m = 50\%$ and the blurry sample ratio $n = 10\%$, and split all classes into 5 learning phases. Following the previous evaluation protocols (Moon et al., 2023; Kang et al., 2025), we report the average any-time accuracy $A_{\text{AUC}}$ and the average last accuracy $A_{\text{Last}}$ as the main metrics. We adopt a ViT-B/16 backbone with Sup-21K, Sup-21/1K, and iBOT-21K checkpoints. The implementation details are included in Appendix B.

**Baselines.** We consider a variety of representative baselines, categorized into three groups: (1) Simple lower-bound methods such as sequential fine-tuning (Seq FT) of the entire model, Seq FT with slow learner (SL) (Zhang et al., 2023) that selectively reduces the backbone learning rate, and linear probing of the fixed backbone. (2) PTMs-based CL methods such as L2P (Wang et al., 2022b), DualPrompt (Wang et al., 2022a), and CODA-P (Smith et al., 2023). Here we follow the previous work (Smith et al., 2023) to re-implement L2P (Wang et al., 2022b) by replacing its prompt tuning with prefix tuning, denoted as Deep L2P, for comparison fairness and the ease of combination with MePo. (3) PTMs-based GCL methods such as MVP (Moon et al., 2023) and MISA (Kang et al., 2025). All PTMs-based methods employ prefix tuning with prompt length 5, inserted into layers 1-5.

**Training Details** We follow the previous implementations (Moon et al., 2023; Kang et al., 2025) to ensure fairness of the comparison. We adopt a ViT-B/16 backbone and consider three ImageNet-21K pretrained checkpoints with different levels of supervision: Sup-21K (`vit-base-patch16-224`) performs supervised pretraining on ImageNet-21K, Sup-21/1K (Ridnik et al., 2021; Dosovitskiy et al., 2020b) performs self-supervised pretraining on ImageNet-21K and supervised finetuning on ImageNet-1K, while iBOT-21K (Zhou et al., 2021) performs self-supervised pretraining on ImageNet-21K. To implement MePo, both $\mathcal{D}_{\text{meta}}$ and $\mathcal{D}_{\text{ref}}$ are constructed from ImageNet-1K (Russakovsky et al., 2015). In MePo Phase I, we construct $\mathcal{D}_{\text{meta}}$ by randomly sampling $|\mathcal{C}_{\text{meta}}| = 100$ classes with 400 samples per class and training-validation split rate $\gamma$

0.3. We employ a SGD optimizer of learning rate $\eta_\theta = 0.0001$ for backbone and learning rate $\eta_\psi = 0.01$ for output layer, and batch size 256 for 50, 10, 150 meta epochs for Sup-21K, Sup-21/1K, iBOT-21K respectively. In MePo Phase II, we construct $\mathcal{D}_{\text{ref}}$ by randomly sampling $|\mathcal{C}| = 1000$ classes with $N_c = 200$ samples per class. To ensure that the Cholesky decomposition remains stable when $\Sigma_{cur}$ is ill-conditioned, we add a small diagonal regularizer (e.g., $\epsilon = 1e - 4$) before decomposition. During the GCL phase, we employ an Adam optimizer of learning rate 0.005 and batch size 64 for 1 epoch.

All the experiments are conducted with one-card 3090 GPU, AMD EPYC 7402 (2.8G Hz).

# C. Additional Results

*Table 4.* Evaluation of hyperparameter $\alpha$ in Eq. (11) with average any-time accuracy $A_{\text{AUC}}(\uparrow)$. We use MVP (Moon et al., 2023) and MISA (Kang et al., 2025) as the baseline implementation. All results are averaged over five runs with different task sequences.

| Setup | MVP w/ MePo | | | | | | MISA w/ MePo | | | | | |
|---|---|---|---|---|---|---|---|---|---|---|---|---|
| | 0 | 0.1 | 0.3 | 0.5 | 0.7 | 1.0 | 0 | 0.1 | 0.3 | 0.5 | 0.7 | 1.0 |
| Sup-21K CIFAR-100 | $71.74_{\pm4.14}$ | $71.86_{\pm4.14}$ | $72.09_{\pm4.26}$ | $72.18_{\pm4.50}$ | $71.63_{\pm4.66}$ | $49.84_{\pm5.82}$ | $81.29_{\pm2.27}$ | $81.59_{\pm2.33}$ | $82.14_{\pm2.56}$ | $82.30_{\pm2.83}$ | $81.56_{\pm3.24}$ | $76.95_{\pm4.48}$ |
| Sup-21K ImageNet-R | $46.32_{\pm1.29}$ | $46.35_{\pm1.31}$ | $46.26_{\pm1.42}$ | $45.76_{\pm1.48}$ | $43.68_{\pm1.34}$ | $34.86_{\pm1.30}$ | $52.35_{\pm2.09}$ | $53.02_{\pm2.06}$ | $54.23_{\pm2.05}$ | $54.86_{\pm2.20}$ | $53.30_{\pm2.23}$ | $39.31_{\pm2.07}$ |
| Sup-21K CUB-200 | $58.67_{\pm2.80}$ | $58.73_{\pm2.93}$ | $58.73_{\pm3.31}$ | $58.08_{\pm3.62}$ | $56.44_{\pm3.86}$ | $42.94_{\pm3.59}$ | $64.83_{\pm2.82}$ | $65.57_{\pm3.04}$ | $66.92_{\pm3.05}$ | $68.13_{\pm3.17}$ | $67.80_{\pm3.27}$ | $63.64_{\pm3.99}$ |
| Sup-21/1K CIFAR-100 | $68.11_{\pm3.93}$ | $68.27_{\pm3.81}$ | $69.00_{\pm3.94}$ | $69.70_{\pm4.01}$ | $70.25_{\pm4.23}$ | $61.57_{\pm5.77}$ | $73.21_{\pm2.16}$ | $73.60_{\pm2.47}$ | $74.59_{\pm2.59}$ | $76.04_{\pm2.85}$ | $78.01_{\pm3.09}$ | $74.87_{\pm4.75}$ |
| Sup-21/1K ImageNet-R | $57.50_{\pm1.18}$ | $57.97_{\pm1.23}$ | $59.05_{\pm1.23}$ | $60.16_{\pm1.19}$ | $61.28_{\pm1.21}$ | $52.54_{\pm1.86}$ | $56.71_{\pm1.08}$ | $57.28_{\pm0.99}$ | $58.95_{\pm0.90}$ | $61.27_{\pm0.99}$ | $64.23_{\pm1.30}$ | $51.97_{\pm1.98}$ |
| Sup-21/1K CUB-200 | $47.26_{\pm3.38}$ | $47.69_{\pm3.42}$ | $48.60_{\pm3.31}$ | $49.40_{\pm3.29}$ | $49.72_{\pm3.53}$ | $35.54_{\pm3.51}$ | $44.68_{\pm2.46}$ | $45.21_{\pm2.52}$ | $46.96_{\pm3.18}$ | $49.23_{\pm3.83}$ | $54.12_{\pm4.09}$ | $55.31_{\pm4.52}$ |
| iBOT-21K CIFAR-100 | $66.53_{\pm4.59}$ | $66.70_{\pm4.73}$ | $66.85_{\pm4.73}$ | $66.88_{\pm4.86}$ | $66.53_{\pm5.14}$ | $60.08_{\pm7.07}$ | $70.52_{\pm2.34}$ | $70.90_{\pm2.48}$ | $72.26_{\pm2.81}$ | $73.83_{\pm3.22}$ | $75.80_{\pm3.77}$ | $66.70_{\pm6.45}$ |
| iBOT-21K ImageNet-R | $52.67_{\pm1.41}$ | $52.92_{\pm1.38}$ | $53.45_{\pm1.40}$ | $53.75_{\pm1.38}$ | $53.36_{\pm1.49}$ | $42.05_{\pm1.95}$ | $50.21_{\pm1.93}$ | $50.85_{\pm2.04}$ | $52.51_{\pm2.17}$ | $54.66_{\pm2.25}$ | $57.00_{\pm2.52}$ | $33.15_{\pm2.11}$ |
| iBOT-21K CUB-200 | $40.61_{\pm3.41}$ | $40.68_{\pm3.38}$ | $40.99_{\pm3.45}$ | $40.76_{\pm3.56}$ | $39.36_{\pm3.72}$ | $25.41_{\pm2.76}$ | $39.44_{\pm2.93}$ | $40.63_{\pm3.28}$ | $42.81_{\pm3.37}$ | $45.81_{\pm3.45}$ | $49.33_{\pm3.59}$ | $39.93_{\pm3.19}$ |

*Table 5.* Evaluation of hyperparameter $\alpha$ in Eq. (11) with average last accuracy $A_{\text{Last}}(\uparrow)$. We use MVP (Moon et al., 2023) and MISA (Kang et al., 2025) as the baseline implementation. All results are averaged over five runs with different task sequences.

| Setup | MVP w/ MePo | | | | | | MISA w/ MePo | | | | | |
|---|---|---|---|---|---|---|---|---|---|---|---|---|
| | 0 | 0.1 | 0.3 | 0.5 | 0.7 | 1.0 | 0 | 0.1 | 0.3 | 0.5 | 0.7 | 1.0 |
| Sup-21K CIFAR-100 | $65.40_{\pm1.99}$ | $66.05_{\pm1.90}$ | $67.47_{\pm1.68}$ | $68.45_{\pm1.59}$ | $68.82_{\pm1.56}$ | $47.43_{\pm2.24}$ | $81.96_{\pm1.12}$ | $82.31_{\pm1.06}$ | $83.18_{\pm1.11}$ | $83.99_{\pm1.35}$ | $84.22_{\pm1.37}$ | $82.06_{\pm1.74}$ |
| Sup-21K ImageNet-R | $38.06_{\pm3.77}$ | $38.21_{\pm3.66}$ | $38.15_{\pm3.61}$ | $37.92_{\pm3.66}$ | $35.91_{\pm4.20}$ | $29.98_{\pm2.43}$ | $45.81_{\pm1.08}$ | $46.55_{\pm1.00}$ | $48.09_{\pm1.00}$ | $49.18_{\pm1.38}$ | $47.78_{\pm1.45}$ | $34.85_{\pm1.06}$ |
| Sup-21K CUB-200 | $51.65_{\pm3.23}$ | $51.69_{\pm3.11}$ | $52.22_{\pm2.80}$ | $52.42_{\pm2.61}$ | $52.36_{\pm2.58}$ | $40.45_{\pm1.74}$ | $59.57_{\pm1.73}$ | $60.08_{\pm1.48}$ | $62.04_{\pm1.43}$ | $64.75_{\pm1.00}$ | $66.72_{\pm0.47}$ | $65.39_{\pm1.59}$ |
| Sup-21/1K CIFAR-100 | $55.88_{\pm3.33}$ | $56.36_{\pm3.31}$ | $58.26_{\pm3.05}$ | $59.43_{\pm3.02}$ | $62.05_{\pm2.39}$ | $60.47_{\pm2.74}$ | $73.21_{\pm2.16}$ | $73.60_{\pm2.47}$ | $74.59_{\pm2.59}$ | $76.04_{\pm2.85}$ | $78.01_{\pm3.09}$ | $74.87_{\pm4.75}$ |
| Sup-21/1K ImageNet-R | $46.75_{\pm4.85}$ | $47.07_{\pm4.71}$ | $48.33_{\pm4.27}$ | $49.53_{\pm4.04}$ | $50.82_{\pm3.70}$ | $46.52_{\pm2.23}$ | $56.71_{\pm1.08}$ | $57.28_{\pm0.99}$ | $58.95_{\pm0.90}$ | $61.27_{\pm0.99}$ | $64.23_{\pm1.30}$ | $51.97_{\pm1.98}$ |
| Sup-21/1K CUB-200 | $39.65_{\pm8.09}$ | $40.44_{\pm7.83}$ | $41.65_{\pm7.71}$ | $42.10_{\pm7.74}$ | $42.81_{\pm6.74}$ | $32.91_{\pm2.81}$ | $44.68_{\pm2.46}$ | $45.21_{\pm2.52}$ | $46.96_{\pm3.18}$ | $49.23_{\pm3.83}$ | $54.12_{\pm4.09}$ | $55.31_{\pm4.52}$ |
| iBOT-21K CIFAR-100 | $55.44_{\pm3.75}$ | $56.22_{\pm3.52}$ | $56.67_{\pm2.85}$ | $57.19_{\pm2.63}$ | $58.26_{\pm1.81}$ | $62.68_{\pm3.11}$ | $70.47_{\pm2.45}$ | $70.78_{\pm2.37}$ | $72.08_{\pm1.21}$ | $73.55_{\pm0.50}$ | $76.02_{\pm1.18}$ | $72.50_{\pm1.95}$ |
| iBOT-21K ImageNet-R | $41.91_{\pm3.95}$ | $42.05_{\pm3.80}$ | $42.39_{\pm3.42}$ | $42.55_{\pm3.08}$ | $42.48_{\pm3.05}$ | $35.44_{\pm3.04}$ | $43.52_{\pm1.39}$ | $43.94_{\pm1.47}$ | $45.17_{\pm1.26}$ | $47.33_{\pm1.32}$ | $49.86_{\pm1.22}$ | $28.91_{\pm0.69}$ |
| iBOT-21K CUB-200 | $34.17_{\pm9.09}$ | $34.54_{\pm9.27}$ | $34.66_{\pm8.40}$ | $34.72_{\pm8.11}$ | $33.89_{\pm7.14}$ | $22.96_{\pm1.31}$ | $40.38_{\pm1.79}$ | $41.02_{\pm2.35}$ | $41.81_{\pm2.19}$ | $43.42_{\pm2.82}$ | $45.68_{\pm2.59}$ | $41.35_{\pm2.80}$ |

*Table 6.* GCL performance with OOD datasets NCH and GTSRB under Sup-21K.

| Method | NCH / Sup-21K | | GTSRB / Sup-21K | |
|---|---|---|---|---|
| | $A_{\text{AUC}}(\uparrow)$ | $A_{\text{Last}}(\uparrow)$ | $A_{\text{AUC}}(\uparrow)$ | $A_{\text{Last}}(\uparrow)$ |
| DualPrompt | 53.36 | 34.39 | 32.24 | 19.46 |
| w/ MePo (Ours) | **55.01** | **37.96** | 32.04 | **20.67** |
| MISA | 69.72 | 53.52 | 56.46 | 39.86 |
| w/ MePo (Ours) | **71.97** | **55.31** | **56.96** | **42.47** |

*Table 7.* Performance of L2P (Wang et al., 2022b) and DualPrompt (Wang et al., 2022a) with and without MePo under different continual learning settings.

| Setting | Method | $A_{Avg}(\uparrow)$ | $A_{Last}(\uparrow)$ | Forgetting ($\downarrow$) |
|---|---|---|---|---|
| Offline CL (Sup-21K CIFAR-100) | L2P | 81.71 | 76.35 | 6.50 |
| | w/ MePo (Ours) | **86.66** | **81.47** | **5.70** |
| | DualPrompt | 88.22 | 83.59 | 4.99 |
| | w/ MePo (Ours) | **89.36** | **84.50** | **4.78** |
| Online CL (Sup-21K CIFAR-100) | L2P | 76.72 | 69.00 | 8.70 |
| | w/ MePo (Ours) | **83.64** | **76.98** | **7.30** |
| | DualPrompt | 81.36 | 76.47 | 6.04 |
| | w/ MePo (Ours) | **85.28** | **80.11** | **5.89** |
| Domain CL (Sup-21K Core50) | L2P | 94.27 | 93.70 | 0.49 |
| | w/ MePo (Ours) | **95.87** | **95.42** | **0.33** |
| | DualPrompt | 96.49 | 96.11 | 0.29 |
| | w/ MePo (Ours) | **97.12** | **96.97** | **0.15** |
| Domain CL (Sup-21K DomainNet) | L2P | 45.69 | 37.18 | 8.11 |
| | w/ MePo (Ours) | **47.87** | **39.81** | **7.98** |
| | DualPrompt | 52.24 | 44.34 | 7.49 |
| | w/ MePo (Ours) | **53.52** | **45.36** | **7.43** |

*Table 8.* Performance using different batch sizes on CIFAR-100 under Sup-21K. All results are averaged over five runs.

| Batch Size | Method | $A_{AUC}(\uparrow)$ | $A_{Last}(\uparrow)$ | Forgetting ($\downarrow$) |
|---|---|---|---|---|
| 10 | L2P | 70.87 | 72.23 | 11.67 |
| | w/ MePo (Ours) | 80.41 | 82.40 | 6.63 |
| | Improvement | **+9.54** | **+10.17** | **-5.04** |
| | MISA | 75.29 | 75.83 | 9.30 |
| | w/ MePo (Ours) | 82.04 | 82.95 | 7.69 |
| | Improvement | **+6.75** | **+7.12** | **-1.61** |
| 32 | L2P | 75.14 | 76.29 | 10.82 |
| | w/ MePo (Ours) | 82.17 | 83.51 | 7.18 |
| | Improvement | **+7.03** | **+7.22** | **-3.64** |
| | MISA | 79.19 | 79.28 | 9.68 |
| | w/ MePo (Ours) | 82.04 | 82.95 | 7.69 |
| | Improvement | **+2.85** | **+3.67** | **-1.99** |
| 64 | L2P | 78.12 | 77.73 | 12.41 |
| | w/ MePo (Ours) | 83.63 | 83.98 | 8.62 |
| | Improvement | **+5.51** | **+6.25** | **-3.79** |
| | MISA | 80.35 | 80.75 | 9.67 |
| | w/ MePo (Ours) | 82.30 | 83.99 | 7.66 |
| | Improvement | **+1.95** | **+3.24** | **-2.01** |

*Table 9.* Effect of different pretraining data on downstream GCL performance using L2P and MISA.

| Method | Pretraining Data | Downstream GCL Data | $A_{AUC}(\uparrow)$ | $A_{LAST}(\uparrow)$ |
|---|---|---|---|---|
| **L2P-based Methods** | | | | |
| w/o MePo | - | ImageNet-R | 42.39 | 38.16 |
| w/ MePo (Ours) | CIFAR-100 | ImageNet-R | **50.72** | **48.40** |
| w/ MePo (Ours) | ImageNet-1K | ImageNet-R | **58.71** | **55.13** |
| w/o MePo | - | CUB200 | 60.95 | 56.31 |
| w/ MePo (Ours) | CIFAR-100 | CUB200 | **63.57** | **64.40** |
| w/ MePo (Ours) | ImageNet-1K | CUB200 | **64.92** | **63.30** |
| **MISA-based Methods** | | | | |
| w/o MePo | - | ImageNet-R | 51.52 | 45.08 |
| w/ MePo (Ours) | CIFAR-100 | ImageNet-R | **54.92** | **49.33** |
| w/ MePo (Ours) | ImageNet-1K | ImageNet-R | **54.86** | **49.18** |
| w/o MePo | - | CUB200 | 65.40 | 60.20 |
| w/ MePo (Ours) | CIFAR-100 | CUB200 | **68.54** | **65.11** |
| w/ MePo (Ours) | ImageNet-1K | CUB200 | **68.13** | **64.75** |

*Table 10.* Effect of varying the number of pseudo-tasks ($T'$) under different GCL task sequence lengths ($T$). Results are averaged over 5 runs using MISA (Kang et al., 2025) on ImageNet-R under Sup-21K.

| Method | Downstream GCL task $T = 5$ | | Downstream GCL task $T = 20$ | |
|---|---|---|---|---|
| | $A_{\text{AUC}}(\uparrow)$ | $A_{\text{Last}}(\uparrow)$ | $A_{\text{AUC}}(\uparrow)$ | $A_{\text{Last}}(\uparrow)$ |
| Baseline (w/o MePo) | 51.49±2.04 | 45.04±1.40 | 48.94±0.62 | 47.88±1.28 |
| Pseudo tasks ($T' = 5$) | 54.15±2.37 | 48.60±1.60 | 50.63±0.92 | 51.49±0.97 |
| Pseudo tasks ($T' = 10$) | 54.31±2.34 | 48.68±1.65 | 50.90±1.00 | 51.62±1.15 |
| Pseudo tasks ($T' = 20$) | 54.41±2.27 | 48.74±1.59 | 51.14±0.94 | 51.70±1.39 |
| Pseudo tasks ($T' = 50$) | **54.55±2.27** | **48.81±1.61** | **51.54±0.92** | **52.10±1.07** |

*Table 11.* Runtime analysis of the meta-refinement phase. Only 50 meta-epochs are required for Sup-21K, and just 10 meta-epochs for Sup-21/1K to reach convergence. Trainable parameters is 87.16M and training time measured on single NVIDIA RTX 3090 GPUs, AMD EPYC 7402 (2.8GHz).

| Meta epoch | 1 | 2 | 3 | 4 | 5 | **Average** |
|---|---|---|---|---|---|---|
| Time (mins) | 14.43 | 14.20 | 14.13 | 14.00 | 13.96 | 14.14 |

*Table 12.* Comparison of data budgets between Sup-21K pretraining and MePo meta-refinement. Sup-21K pretraining on ImageNet-21K for 90 epochs processes ~1.3B images, while MePo meta-refinement on ImageNet-1K (400 samples per class) for 50 epochs processes only ~2M images (0.15% of pretraining).

| Training | Image Size | Epochs | Batch Size | Total Steps | Images Processed |
|---|---|---|---|---|---|
| Sup-21K (Pretraining) (Dosovitskiy et al., 2020a) | 224×224 | 90 | 4096 | ~310k | ~1.3B |
| Sup-21K (MePo Post-Refinement) | 224×224 | 50 | 256 | ~12.8k | ~2M |

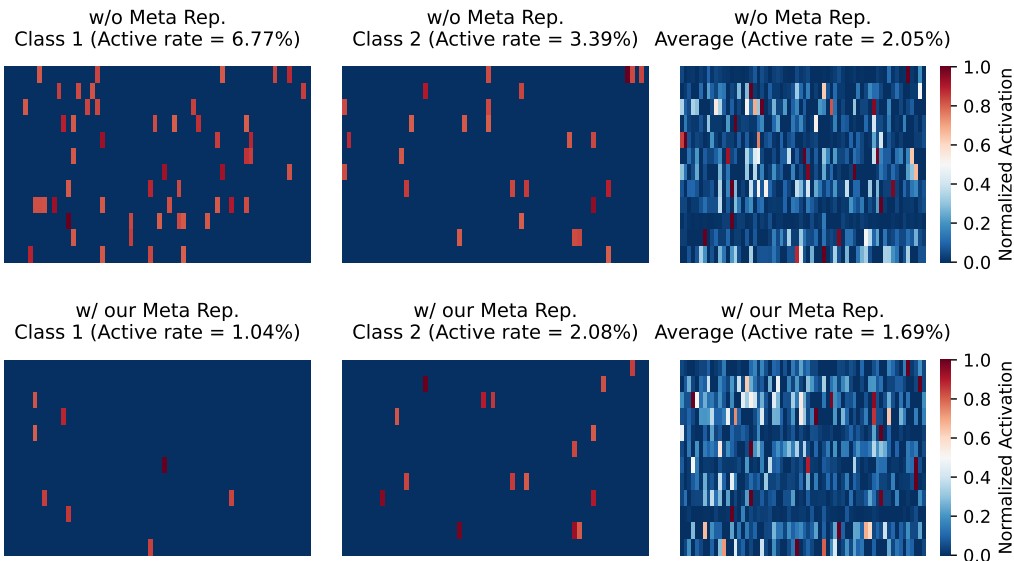

*Figure 8.* Visualization of feature representation with Meta Rep. We reshape the 768 length class-prototype representation vectors into 12x64, normalize and visualize them with threshold 0.8; here random class means representation for a randomly chosen class-prototype from ImageNet-R, whereas average activation is the mean representation for the all classes.

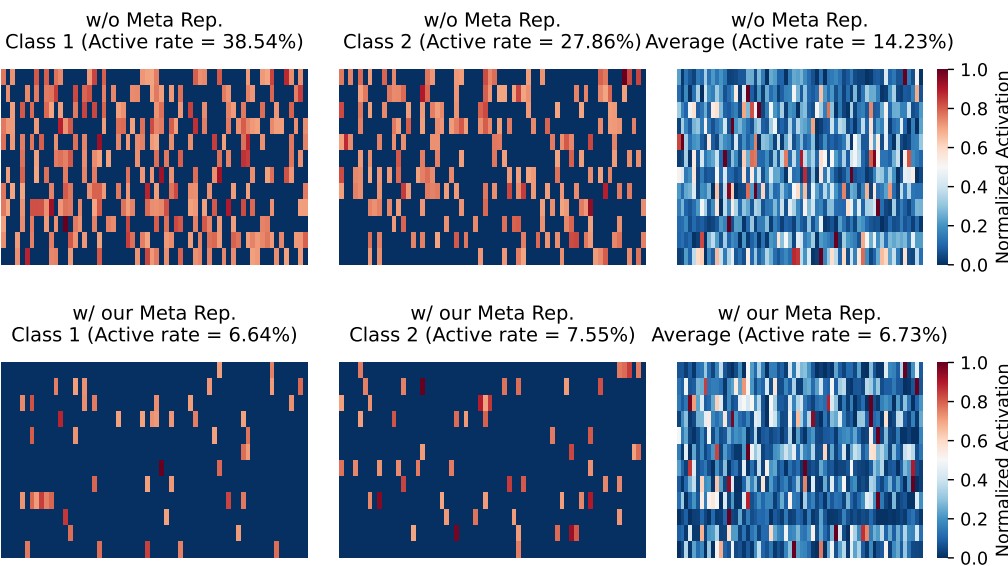

*Figure 9.* Visualization of feature representation with Meta Rep. We reshape the 768 length class-prototype representation vectors into 12x64, normalize and visualize them with threshold 0.7; here random class means representation for a randomly chosen class-prototype from ImageNet-R, whereas average activation is the mean representation for the all classes.

---

**Algorithm 1** Meta Post-Refinement (MePo) for General Continual Learning (GCL)

---

1: **Input**: Pretraining dataset $\mathcal{D}_{\text{pre}}$, meta-learning rate $\eta_{\text{meta}}$
2: **Hyperparameters**: Meta-epochs $K$, tasks per meta-epoch $T'$, learning rates $\eta_\theta, \eta_\psi$, weight $\alpha$
3:
4: **Step 1: Meta-Learning for Representation Learning**
5: Initialize backbone $\theta^{(0)} \leftarrow$ pretrained parameters
6: **for** meta-epoch $k = 1$ to $K$ **do**
7: $\quad \triangleright$ *Construct pseudo task sequence*
8: $\quad$ Sample $\mathcal{D}_{\text{meta}}^{(k)} \subset \mathcal{D}_{\text{pre}}$ with $\mathcal{C}_{\text{meta}}$ classes
9: $\quad$ Split $\mathcal{D}_{\text{meta}}^{(k)}$ into $\{\hat{\mathcal{D}}_t^{(k)}\}_{t=1}^{T'}$ (sequential training) and $\mathcal{D}_{\text{joint}}^{(k)}$ (joint training)
10: $\quad \triangleright$ *Inner loop: sequential training*
11: $\quad$ Initialize $\theta_0^{(k-1)} \leftarrow \theta^{(k-1)}, \psi_0 \leftarrow$ random
12: $\quad$ **for** task $t = 1$ to $T'$ **do**
13: $\quad\quad$ Compute $\mathcal{L}_t$ via Eq.(2) on $\hat{\mathcal{D}}_t^{(k)}$
14: $\quad\quad$ Update $\theta_t^{(k-1)} \leftarrow \theta_{t-1}^{(k-1)} - \eta_\theta \nabla_\theta \mathcal{L}_t$ $\hfill \triangleright$ Eq.(3)
15: $\quad\quad$ Update $\psi_t \leftarrow \psi_{t-1} - \eta_\psi \nabla_\psi \mathcal{L}_t$
16: $\quad$ **end for**
17: $\quad \triangleright$ *Outer loop: joint training*
18: $\quad$ Refine $\hat{\theta}_{T'}^{(k-1)}$ via Eq.(5) on $\mathcal{D}_{\text{joint}}^{(k)}$
19: $\quad \triangleright$ *Meta-parameter accumulation*
20: $\quad$ Update $\theta^{(k)} \leftarrow \theta^{(k-1)} + \eta_{\text{meta}}(\hat{\theta}_{T'}^{(k-1)} - \theta^{(k-1)})$ $\hfill \triangleright$ Eq.(6)
21: **end for**
22: Obtain optimized backbone $\theta^* \leftarrow \theta^{(K)}$
23:
24: **Step 2: Meta Covariance Initialization**
25: Sample reference data $\mathcal{D}_{\text{ref}} \subset \mathcal{D}_{\text{pre}}$ with $C_{\text{ref}}$ classes
26: Compute class prototypes $\{\boldsymbol{\mu}_c\}$ via Eq.(7)
27: Calculate $\boldsymbol{\Sigma}_{\text{pre}} \leftarrow \frac{1}{\mathcal{C}_{\text{ref}}-1} \sum_c (\boldsymbol{\mu}_c - \bar{\boldsymbol{\mu}})(\boldsymbol{\mu}_c - \bar{\boldsymbol{\mu}})^\top$ $\hfill \triangleright$ Eq.(8)
28:
29: **Step 3: Feature Alignment in GCL**
30: **for** each incoming batch $\mathcal{B}$ in GCL tasks **do**
31: $\quad \triangleright$ *Current feature statistics*
32: $\quad$ Compute $\boldsymbol{\Sigma}_{\text{cur}}$ via Eq.(9)
33: $\quad$ Decompose $\boldsymbol{\Sigma}_{\text{cur}} = \boldsymbol{L}_{\text{cur}}\boldsymbol{L}_{\text{cur}}^\top$, $\boldsymbol{\Sigma}_{\text{pre}} = \boldsymbol{L}_{\text{pre}}\boldsymbol{L}_{\text{pre}}^\top$
34: $\quad \triangleright$ *Feature transformation*
35: $\quad$ Compute $\boldsymbol{A} \leftarrow \boldsymbol{L}_{\text{cur}}^{-1}\boldsymbol{L}_{\text{pre}}$
36: $\quad$ **for** each feature $\mathbf{f}_i \in \mathcal{B}$ **do**
37: $\quad\quad \hat{\mathbf{f}}_i \leftarrow \mathbf{f}_i \boldsymbol{A}$ $\hfill \triangleright$ Eq.(10)
38: $\quad\quad \mathbf{f}_{\text{trans},i} \leftarrow \alpha\hat{\mathbf{f}}_i + (1-\alpha)\mathbf{f}_i$ $\hfill \triangleright$ Eq.(11)
39: $\quad$ **end for**
40: $\quad \triangleright$ *Model update*
41: $\quad$ Update $\Delta\theta, \psi$ via $\mathcal{L}_{\text{CE}}$ on $\{\mathbf{f}_{\text{trans},i}\}_{i=1}^{|\mathcal{B}|}$ $\hfill \triangleright$ Eq. (12)
42: **end for**
43: **Return** Adapted backbone $\theta^* + \Delta\theta$, aligned classifier $\psi$

---

# D. Theoretical Analysis of Meta-Rep

We provide a theoretical justification for why the Meta-Rep of MePo improves continual learning (CL) performance. Specifically, we show that the meta-learned initialization reduces gradient interference and better aligns sequential updates with joint updates, thereby mitigating catastrophic forgetting.

## D.1. Setup and Notation

Let $\theta \in \mathbb{R}^d$ denote the backbone parameters of a neural network before learning a new task. Meta-Rep optimizes $\theta$ via meta-learning over pseudo-task sequences sampled from the pretraining data.

Let a pseudo-task sequence be denoted by $\mathcal{T} = (\hat{\mathcal{D}}_{1:T'}, \mathcal{D}_{\text{joint}})$, where $\hat{\mathcal{D}}_1, \dots, \hat{\mathcal{D}}_{T'}$ form a sequential training stream, and $\mathcal{D}_{\text{joint}}$ is a held-out validation set containing all classes in the sequence.

Each loss $L_t(\theta)$ denotes the empirical loss over task $t$ from $\hat{\mathcal{D}}_t$, and $L_{\text{joint}}(\theta)$ is the joint loss over $\mathcal{D}_{\text{joint}}$.

The inner-loop update of Meta-Rep at time $t$ is:

$$\theta_t = \theta_{t-1} - \eta_\theta \nabla L_t(\theta_{t-1}), \qquad t = 1, \dots, T', \tag{13}$$

followed by a meta-validation step:

$$\theta_{T'+1} = \theta_{T'} - \eta_\theta \nabla L_{\text{joint}}(\theta_{T'}), \tag{14}$$

where $\eta_\theta$ is the learning rate.

We define the overall inner-loop operator as:

$$F(\theta, \mathcal{T}) := \theta_{T'+1}. \tag{15}$$

Then, Meta-Rep applies a Reptile-style (Nichol & Schulman, 2018) meta-update:

$$\theta' = \theta + \eta_\theta \left( F(\theta, \mathcal{T}) - \theta \right), \tag{16}$$

## D.2. Main Result

We formally characterize how Meta-Rep reduces the deviation between sequential and joint updates, which serves as a surrogate for mitigating forgetting.

**Theorem D.1** (Sequential Update Consistency Theorem). *Let $\theta^\star$ be a stationary point of the surrogate meta-objective:*

$$\tilde{J}(\theta) = \mathbb{E}_\mathcal{T} \left[ \sum_{t=1}^{T'} L_t(\theta) + L_{joint}(\theta) \right], \tag{17}$$

*obtained by applying the meta-update in* (16). *Assume each loss $L_t$ and $L_{joint}$ is twice differentiable, with bounded gradients and Hessians, and the inner-loop step size $\eta$ is sufficiently small.*

*Then for any two-task sequence $A \to B$ and some constant $C > 0$, the deviation between sequential and joint updates satisfies:*

$$\|\theta_{seq} - \theta_{joint}\| \leq C \cdot \eta^2 \|H_B(\theta^\star)\| \cdot \|\nabla L_A(\theta^\star)\| + \mathcal{O}(\eta^3),$$

*Moreover, if the pseudo-task distribution approximates the downstream continual learning distribution, this bound is strictly smaller than the same quantity evaluated at a generic pretrained initialization $\theta_0$:*

$$\|\theta_{seq} - \theta_{joint}\|\Big|_{\theta^\star} < \|\theta_{seq} - \theta_{joint}\|\Big|_{\theta_0}.$$

## D.3. Proof of Theorem

We proceed to prove Theorem D.1. The proof is organized in the following steps.

**Step 1: Reptile Expansion.** We apply Taylor expansion to each gradient update in the inner loop. For any $t$, we have:

$$\nabla L_t(\theta_{t-1}) = \nabla L_t(\theta) + \nabla^2 L_t(\tilde{\theta}_t)(\theta_{t-1} - \theta), \tag{18}$$

for some $\tilde{\theta}_t$ on the line between $\theta_{t-1}$ and $\theta$. Since each $\theta_{t-1} - \theta = \mathcal{O}(\eta_\theta)$, we get:

$$\nabla L_t(\theta_{t-1}) = \nabla L_t(\theta) + \mathcal{O}(\eta_\theta). \tag{19}$$

Substituting into the inner-loop update and unrolling over all tasks:

$$F(\theta, \mathcal{T}) = \theta - \eta_\theta \sum_{t=1}^{T'} \nabla L_t(\theta) - \eta_\theta \nabla L_{\text{joint}}(\theta) + \mathcal{O}(\eta_\theta^2). \tag{20}$$

Taking expectation over pseudo-tasks:

$$\mathbb{E}_{\mathcal{T}}\left[F(\theta, \mathcal{T}) - \theta\right] = -\eta_\theta \nabla \tilde{J}(\theta) + \mathcal{O}(\eta_\theta^2). \tag{21}$$

Under Robbins–Monro conditions (Robbins & Monro, 1951) on the meta step size (diminishing, square-summable), this ensures convergence to a stationary point $\theta^\star$ of $\tilde{J}(\theta)$.

**Step 2: Forgetting Gap Between Sequential and Joint Updates.** For a two-task sequence $A \to B$, define:

$$\theta_{\text{joint}} = \theta - \eta\left(\nabla L_A(\theta) + \nabla L_B(\theta)\right), \tag{22}$$
$$\theta_{\text{seq}} = \theta - \eta \nabla L_A(\theta) - \eta \nabla L_B(\theta - \eta \nabla L_A(\theta)). \tag{23}$$

Taylor expanding $\nabla L_B(\cdot)$ around $\theta$:

$$\nabla L_B(\theta - \eta \nabla L_A(\theta)) = \nabla L_B(\theta) - \eta H_B(\theta) \nabla L_A(\theta) + \mathcal{O}(\eta^2). \tag{24}$$

Substituting into (23), we obtain:

$$\theta_{\text{seq}} - \theta_{\text{joint}} = \eta^2 H_B(\theta) \nabla L_A(\theta) + \mathcal{O}(\eta^3). \tag{25}$$

Therefore, the forgetting error satisfies:

$$\|\theta_{\text{seq}} - \theta_{\text{joint}}\| \leq C \cdot \eta^2 \|H_B(\theta)\| \cdot \|\nabla L_A(\theta)\| + \mathcal{O}(\eta^3). \tag{26}$$

**Step 3: Effect of Minimizing $\tilde{J}$.** Both $\|\nabla L_A(\theta)\|$ and $\|H_B(\theta)\|$ appear in the meta-objective $\tilde{J}(\theta)$. Hence, minimizing $\tilde{J}$ at $\theta^\star$ reduces both terms:

$$\|H_B(\theta^\star) \nabla L_A(\theta^\star)\| \leq \|H_B(\theta^\star)\| \cdot \|\nabla L_A(\theta^\star)\| \downarrow. \tag{27}$$

Combining with (26) shows that:

$$\|\theta_{\text{seq}} - \theta_{\text{joint}}\| \text{ is minimized at } \theta^\star,$$

concluding the proof.

### D.4. Interpretation

The Meta-Rep update approximates first-order gradient descent on a surrogate meta-objective $\tilde{J}(\theta)$, which integrates both sequential learning loss and joint loss over pseudo-tasks. This objective implicitly encourages smoother loss landscapes (via small Hessians) and more stable gradients. As a result, the update discrepancy between sequential and joint training is reduced, which improves model stability and mitigates catastrophic forgetting. These theoretical insights align with our empirical observations in Sec. 4.

