# OpenReview forum: "MePo: Meta Post-Refinement for Rehearsal-Free General Continual Learning"
_ICML.cc/2026/Conference — ICML 2026 regular_

### Official Review · Reviewer_MBU4 · 2026-03-09

**Soundness:** 3
**Presentation:** 3
**Significance:** 3
**Originality:** 3
**Overall Recommendation:** 5
**Confidence:** 3

**Summary:**

MePo (Meta PostRefinement) is a novel framework designed to improve general continual learning (GCL) with pretrained models (PTMs). The core innovation is a bi-level meta-learning approach that refines pretrained backbones before deployment on downstream GCL tasks. Rather than relying solely on upstream pretraining, MePo adds a post-refinement phase using pseudo task sequences constructed from pretraining data. This serves as an extended pretraining step that enhances the model's ability to rapidly adapt to online, temporally mixed data streams without task boundaries. Additionally, MePo initializes a meta covariance matrix to capture the reference geometry of the pretrained representation space, enabling robust output alignment through second-order statistics. The method operates in a rehearsal-free manner (without storing past data) and achieves substantial performance improvements across multiple benchmarks—for example, 15.10% improvement on CIFAR-100, 13.36% on ImageNet-R, and 12.56% on CUB-200.

**Compliance With Llm Reviewing Policy:**

Affirmed.

**Final Justification:**

Through extensive ablations and cross-domain evaluations, we have demonstrated that MePo is robust, data-efficient, and broadly generalizable, confirming its practical value as a one-time, shareable meta-refinement strategy for generalized continual learning.

**Key Questions For Authors:**

* How sensitive is MePo to the quality and diversity of pretraining data used for constructing pseudo tasks?
* Does MePo require task-aware pseudo task construction, or can it work with purely random task splits of pretraining data?
* How does the method perform when the pretraining data distribution is significantly different from downstream GCL tasks?
* Can the meta-refinement phase be shared across multiple downstream GCL benchmarks, or does it need to be recomputed per benchmark?

**Limitations:**

* Limited algorithmic novelty: Meta-learning, pseudo tasks, and covariance-based alignment are all established techniques. The originality lies in their application combination rather than fundamental innovation.
* Pseudo task construction is not novel: The paper doesn't explain whether this is a new technique or a standard approach (e.g., random task splits of pretraining data). If standard, the novelty is further reduced.
* Covariance-based alignment is well-known: Using second-order statistics for feature alignment or regularization has been explored in prior work. The specific application to GCL output alignment may be novel, but the core technique is not.
* Incremental over existing PTM-based CL methods: The paper builds directly on MVP and MISA; MePo is essentially a plug-in enhancement rather than a fundamentally different approach.
* No new theoretical insights: The paper doesn't introduce new concepts or theory; it applies existing techniques in a new context.

**Strengths And Weaknesses:**

Strengths
* Well-motivated biological inspiration: The connection to meta-plasticity and reconstructive memory from neuroscience provides principled grounding for the approach, moving beyond purely engineering-driven solutions.
* Addresses a genuine gap: The paper identifies a real limitation in existing PTMs-based CL methods—they struggle with online datastreams and blurry task boundaries (GCL) because they assume disjoint task boundaries. MePo tackles this practical challenge head-on.
* Novel two-pronged contribution: The combination of (1) meta-learned pseudo task sequences for backbone refinement and (2) meta-covariance initialization for output alignment is creative and appears to be the first of its kind in this domain.
* Plug-in versatility: MePo works across different PTM checkpoints and GCL benchmarks without requiring rehearsal (memory of past data), making it practical and generalizable.
* Substantial empirical gains: The reported improvements (12–15% across major benchmarks) are significant and suggest the method is not marginal.

Weaknesses
* Pseudo task construction still opaque: Even with experimental results, the paper doesn't clearly explain how pseudo task sequences are constructed from pretraining data. What defines a pseudo task? How many are used? How are they sampled? This is critical for reproducibility.
* Inconsistent gains across baselines: Table 3 shows that improvements from Meta Rep and Meta Cov vary significantly depending on the baseline method and dataset. For instance, some configurations show marginal or inconsistent gains, suggesting the method may not generalize uniformly or that some baselines are better suited to MePo than others.
* Self-supervised PTM analysis incomplete: While the paper emphasizes that self-supervised PTMs are "more realistic yet often underfitted," there's no detailed analysis of why MePo specifically addresses this underfitting. Is it the meta-refinement, the covariance alignment, or both?
* Missing ablation on pseudo task design: There's no ablation study on how the number, diversity, or construction strategy of pseudo tasks affects performance. This is a core component of the method that deserves investigation.
* Comparison with simpler post-refinement strategies: The paper doesn't compare MePo against simpler alternatives, such as standard fine-tuning on pretraining data or other post-hoc refinement approaches. This would help isolate the value of the meta-learning framework specifically.
* Limited discussion of when MePo is most beneficial: While the paper shows MePo helps more on harder datasets, there's no clear guidance on which characteristics of a downstream task or PTM would indicate whether MePo is likely to help significantly.

---

> ### Author Rebuttal · Authors · 2026-03-30
>
> We sincerely thanks for your positive recognition of MePo. Below, we provide a point-to-point response to your questions, and will incorporate all additions in the final version.
>
> **W1 & Q1 & Q2: Construction of the pseudo-tasks**
>
> As detailed in Appendix B, we provide the training details of MePo. To further demonstrate the robustness and practicality of MePo, we conducted extensive ablation studies on the key hyperparameters during the offline meta-refinement phase. The results indicate that MePo is remarkably robust, achieving significant performance gains over the baselines without requiring heavy hyperparameter tuning.
>
> As shown in the Table 9 and Table 10, we have validated MePo is insensitive of  **Number of pseudo-tasks** and **Source of pretraining data**.
>
> **Learning rate**: The meta-refinement process exhibits stable convergence across a wide range of learning rates. While a higher learning rate yields slightly better metrics, all tested rates from 0.05 to 0.5 safely and consistently outperform the unrefined baseline.
>
> |Learning rate|w/ ours|0.05|0.1|0.5|
> |-|-|-|-|-|
> |$A_{Auc}$|51.52±2.09|54.65±2.28|54.86±2.20|55.56±2.00|
> |$A_{Last}$|45.08±1.43|48.89±1.47|49.18±1.38|49.50±0.75|
>
> **Data efficiency**: Using as few as 50 samples per class nearly matches the performance of using 200 samples and vastly outperforms the unrefined baseline. Increasing the data budget yields only marginal diminishing returns, proving that MePo does not rely on massive data availability.
>
> |Samples per class|w/ ours|50|100|200|
> |-|-|-|-|-|
> |$A_{Auc}$|51.52±2.09|54.48±2.29|54.55±2.26|54.86±2.20|
> |$A_{Last}$|45.08±1.43|48.67±1.50|48.82±1.53|49.18±1.38|
>
> **Q3: Domain mismatch**
>
> To address concerns regarding domain mismatch, we evaluated MePo on benchmarks with **strong distribution shifts** (Domain CL seetting, see Table 7) and **distinct out-of-distribution semantics** (NCH chest X-rays, GTSRB traffic signs, see Table 6). The results demonstrate that MePo consistently improves baseline performance across these diverse scenarios, confirming its robust transferability and generalizability far beyond ImageNet-like domains.
>
> **Q4: Sharing the meta-refinement phase**
>
> Meta-refinement phase is only **one-time** and can be shared across multiple downstream GCL benchmarks without recomputing.
>
> **W5: Comparison with simpler post-refinement strategies**
>
> To further validate this, we compare the downstream GCL performance of MISA using a standard fine-tuned backbone versus our MePo backbone, with both strictly constrained to identical data budgets. MePo consistently outperforms standard fine-tuning across all evaluated datasets. This confirms that our performance gains stem specifically from the bi-level meta-objective rather than from mere exposure to the pretraining data.
>
> ||CIFAR-100||ImageNet-R||CUB-200||
> |-|-|-|-|-|-|-|
> ||$A_{Auc}$|$A_{Last}$|$A_{Auc}$|$A_{Last}$|$A_{Auc}$|$A_{Last}$|
> |w/ Standard Fine-Tuning|80.35±2.40|80.80±1.14|51.53±2.07|45.11±1.34|65.39±2.96|60.15±1.94|
> |w/ MePo|**82.30±2.83**|**83.99±1.35**|**54.86±2.20**|**49.18±1.38**|**68.13±3.17**|**64.75±1.00**|
>
> **W3 & W6: When is MePo most beneficial? (Self-supervised analysis)**
>
> MePo is most beneficial when the underlying PTM lacks separability and adaptability, a common trait in self-supervised models (like iBOT or Sup-21/1K). Self-supervised learning often coverges to flatter, highly entangled local minima [1-3], which are robust but exceptionally difficult to adapt via online, single-pass gradients. MePo specifically addresses this by enforcing sparse activations (Fig. 7) and aligning geometric distortions (Fig. 6), making these stubborn self-supervised representations highly malleable for GCL.
>
> The impact of SSL on CL operates at two levels in PTMs. SSL is beneficial mainly during the continual pretraining stage, where contrastive objectives produce flatter minima and robust representations [1-3]. However, self-supervised PTMs often perform worse in downstream CL (continual fine-tuning), especially under parameter-efficient tuning [4-5] (Table 1), because their learned representations are ``too robust'' and thus more difficult to adapt to specific tasks. Although prior work reports strong SSL performance in traditional CL, our results show a different pattern in GCL: self-supervised PTMs such as iBOT lag behind supervised PTMs (Sup-21K) without MePo (Table 1, MISA: 80.35 vs. 65.30 on CIFAR-100). We hypothesize that SSL yields less linearly separable features, making single-pass online adaptation difficult. MePo offers substantial improvements for self-supervised PTMs (65.30 $→$ 75.80) by refining their representations for GCL.
>
> [1] Co2l: Contrastive continual learning, ICCV, 2021
>
> [2] The challenges of continuous self-supervised learning, ECCV, 2022
>
> [3] Self-supervised models are continual learners, CVPR, 2022
>
> [4] Visual prompt tuning, ECCV, 2022
>
> [5] Improving visual prompt tuning for self-supervised vision transformers, ICML, 2023

---

> > ### Author Rebuttal · Reviewer_MBU4 · 2026-04-04
> >
> > The rebuttal thoroughly addresses my concerns with new ablation studies on pseudo-task construction, learning rate sensitivity, data efficiency, domain mismatch evaluation, and a direct comparison against standard fine-tuning. The results are convincing and demonstrate MePo's robustness and practicality. I maintain my original score of 5 as the overall contribution, while solid, remains a combination of existing techniques (meta-learning, pseudo tasks, covariance alignment) without fundamental algorithmic novelty.

---

> > > ### Author Response · Authors · 2026-04-04
> > >
> > > We sincerely thank you for your time and valuable feedback. Please do not hesitate to let us know if you have any further questions.

---

### Official Review · Reviewer_b5CF · 2026-03-10

**Soundness:** 3
**Presentation:** 2
**Significance:** 3
**Originality:** 3
**Overall Recommendation:** 4
**Confidence:** 3

**Summary:**

In this paper, the author examines the general continual learning(GCL) problem. Currently, the GCL problem is shifting toward parameter-efficient tuning for representation learning. However, existing methods usually fail to capture the online data stream. So, the author proposes the MePo, which includes post-refinements via a Meta-learning framework.

In detail, the authors build a pseudo-task for pretraining data and use it as refinement data. And authors use a bi-level train framework the refine the pretrained model. The inner loop will be a sequence of training pseudo tasks. The outer loop will be joint training across all tasks.

For the outer loop, the author proposes adding a meta-covariance matrix to the pretraining data. This matrix can capture the reference geometry for output alignment. In the GCL problem, the input task often has imbalanced class distributions. So the authors first get class-wise features from a subset of pretraining features. In the refinement phase, for each training sample, the authors will first get the feature covariance as pre-aligned features. Then they align the distributions of the current training batch and the reference batch to get post-aligned features as a more balanced feature representation.

**Compliance With Llm Reviewing Policy:**

Affirmed.

**Final Justification:**

The author clearly show the overhead problem is not a major threshold for the whole methods. So they solve one of my major concerns. In this case, I decide to improve my score to 4 as weak-accept.

**Key Questions For Authors:**

1: How does your experiment align with the baseline experiments? Some baselines report better results comapred with the result in this paper.

2: What is the training cost of this framework? The training overhead is also large in meta-learning.

I am willing to improve score if these problems are more clearly addressed.

**Limitations:**

Some overhead problems are not clearly addresed.

**Strengths And Weaknesses:**

Strength:
1: Meta-training on pretrained data for better adaption on continual-learning problems is interesting.
2: The covariance matrix can efficiently address the class-balance problem.
3: The reported experiment result is good on different baselines.

Weakness:
1: The meta-learning process also introduces overhead in the training stage, so it is also important to report the training cost for the framework.
2: The build of pseudo-task is not very clear. The performance can be sensitive to pseudo-task.
3: The experiment setting is not well-aligned with the baseline. The authors do not mention any buffer when several baselines have different buffer settings

---

> ### Author Rebuttal · Authors · 2026-03-30
>
> ## Response to Reviewer b5CF
>
> We sincerely thank you for acknowledging the effectiveness of MePo. Below, we provide a point-to-point response to your questions, and will incorporate all additions in the final version.
>
> **Q1 & W3: Alignment with baseline**
>
> We hereby clarify that the results of main baselines are inherited from prior research [1]. According to the authors [1], these baselines have been optimally tuned for the Si-Blurry benchmarks. Additionally, we conduct **all experiments w/ and w/o MePo under identical hyperparameter configurations** to ensure fair comparison.
>
> In addition, we conducted additional experiments to evaluate MePo in a rehearsal-based setting. We augmented the MISA with memory replay buffers of 500 and 2000 samples, utilizing the Sup-21K backbone on the ImageNet-R benchmark. As shown in the table below, MePo remains highly effective even when memory replay is explicitly introduced, demonstrating the structural knowledge condensed during MePo's offline refinement provides benefits that are highly complementary to standard rehearsal strategies.
>
> |Buffer size|Method|$A_{auc}$|$A_{last}$|
> |-|-|-|-|
> |500|MISA|56.35±1.05|49.67±0.29|
> ||w/ ours|**60.19±1.13**|**54.88±0.36**|
> |2000|MISA|60.03±0.86|56.49±0.66|
> ||w/ ours|**63.41±1.01**|**60.91±0.38**|
>
> [1] Advancing prompt-based methods for replay-independent general continual learning, ICLR, 2025
>
> [2] Dualprompt: Complementary prompting for rehearsal-free continual learning, ECCV, 2022
>
> **W1 & Q2: Overhead of MePo**
>
> We highlight that the computational cost of post-refinement, as an extended pretraining phase, is only **one-time** and is **almost negligiable** compared to the entire pretraining. Thanks to this advance preparation, downstream CL/GCL becomes remarkably **effective and efficient**. Here we provide detailed analyses:
>
> **(1) Post-Refinement (Offline, One-Time Cost):**
>
> Our meta-refinement is a one-time, method-agnostic process: the refined backbone can be directly reused across any downstream CL/GCL tasks without retraining. It is also highly efficient, requiring only 400 images per class from ImageNet-1K. Convergence is reached in just 50 epochs for Sup-21K (or 10 for Sup-21/1K), evaluated on a 3× RTX 3090 GPU setup.
>
> |Meta Epoch|1|2|3|4|5|Average|
> |-|-|-|-|-|-|-|
> |Training time (mins)|14.43|14.20|14.13|14.00|13.96|14.14|
>
> As shown in Table 12, Sup-21K on ImageNt-21K needs to train ~1.3 billion image instances for 90 epochs, but MePo on ImageNt-1K (randomly sampling 400 images per class) only needs to trains additionally ~2 million image instances for 50 epochs, accounting for **only 0.15%** of the entire pretraining.
>
> **(2) GCL (Online, Per-Batch Cost):**
>
> During GCL, the refined backbone ($\theta^*$) is frozen (see Fig. 1). Only lightweight parameters with parameter efficient tuning ($\Delta\theta$) and the output head ($\psi$) are updated. As shown in Table 2, the **GCL phase incurs negligible overhead**, e.g., +0.67% parameters and no additional batch time.
>
> **W2: Construction of the pseudo-tasks**
>
> The pseudo-tasks are constructed purely by randomly splitting a subset of the pretraining data. As detailed in Appendix B, we sample 100 classes with 400 samples per class from ImageNet-1K. These are split into a sequential meta-training set (inner loop) and a joint meta-validation set (outer loop). The fact that MePo achieves state-of-the-art results using simple random splits demonstrates that it is highly robust and does not require complex or sensitive task-engineering.
>
> To further demonstrate the robustness and practicality of MePo, we conducted extensive ablation studies on the key hyperparameters during the offline meta-refinement phase.  The results indicate that MePo is remarkably robust, achieving significant performance gains over the baselines without requiring heavy hyperparameter tuning.
>
> As shown in the Table 9 and Table 10, we have validated MePo is insensitive of **Number of pseudo-tasks** and **Source of pretraining data**.
>
> **Learning rate**: The meta-refinement process exhibits stable convergence across a wide range of learning rates. While a higher learning rate yields slightly better metrics, all tested rates from 0.05 to 0.5 safely and consistently outperform the unrefined baseline.
>
> MISA on Sup-21K/ImageNet-R
> |Learning rate|w/ ours|0.05|0.1|0.5|
> |-|-|-|-|-|
> |$A_{Auc}$|51.52±2.09|54.65±2.28|54.86±2.20|55.56±2.00|
> |$A_{Last}$|45.08±1.43|48.89±1.47|49.18±1.38|49.50±0.75|
>
> **Data efficiency**: Using as few as 50 samples per class nearly matches the performance of using 200 samples and vastly outperforms the unrefined baseline. Increasing the data budget yields only marginal diminishing returns, proving that MePo does not rely on massive data availability.
>
> |Samples per class|w/ ours|50|100|200|
> |-|-|-|-|-|
> |$A_{Auc}$|51.52±2.09|54.48±2.29|54.55±2.26|54.86±2.20|
> |$A_{Last}$|45.08±1.43|48.67±1.50|48.82±1.53|49.18±1.38|

---

> > ### Author Rebuttal · Reviewer_b5CF · 2026-04-03
> >
> > I appreciate the clarification provided by the authors. Especially, the overhead problem has been addressed in the rebuttal, which is my main concern. So I decide to increase the score to 4 as weak-accpet.  I think the author can still improve their final paper by considering the opinions mentioned by all the reviewers.

---

> > > ### Author Response · Authors · 2026-04-04
> > >
> > > Thank you very much for the positive update and for your careful reconsideration of our paper. We are glad that the rebuttal has addressed your main concern regarding the overhead issue, and we also appreciate your suggestion to further improve the final version by considering all reviewers’ feedback.
> > >
> > > We are encouraged by your comment that the score would be updated to **4 (weak accept)**. If convenient, we would greatly appreciate the corresponding update in the system. Thank you again for your time and thoughtful evaluation.

---

### Official Review · Reviewer_MnQA · 2026-03-12

**Soundness:** 3
**Presentation:** 3
**Significance:** 3
**Originality:** 2
**Overall Recommendation:** 4
**Confidence:** 3

**Summary:**

The paper addresses the General Continual Learning (GCL) task and its challenges. GCL is the most general setting among many continual learning setups. This work focuses on GCL methods based on pretrained models (PTM). The paper's main contribution is a plug-in strategy (MePo) for these CL approaches to improve their performance. It is inspired by the meta-learning literature and proposes bi-level optimization, where a pretrained backbone is meta-trained on CL task sequences to refine it into a CL-ready backbone. It also proposes output alignment via a frozen meta-covariance matrix. Its primary contribution is the combination and positioning of these ideas into a coherent "post-refinement" stage. Experiments on CIFAR-100, ImageNet-R, and CUB-200 under the Si-Blurry GCL benchmark show strong performance gains over prior state-of-the-art.

**Compliance With Llm Reviewing Policy:**

Affirmed.

**Final Justification:**

The rebuttal has addressed most of my concerns, that could be hindering acceptance. I maintain my "4: Weak accept" score. While it is valuable addition to literature, the paper's method is still pretty incremental. In the final paper, I hope to see significant reframing of novelties and relation to previous work, as discussed in weaknesses, rebuttal and further comments (including other reviewer's concerns). In this case, I would support the acceptance.

**Key Questions For Authors:**

Q1) Is there anything structurally different in the bi-level formulation beyond the fact that task construction is GCL-specific and aimed at PTMs instead of training from scratch?

Q2) Why did MePo not improve MVP as much as it did for MISA?

Q3) With MePo, DualPrompt performs worse (A_last for CIFAR-100, ImageNet-R, CUB200 under Sup-21/1K). Why is that?

Q4) When the downstream domain is far from the pretraining domain (e.g., table 6 - NCH, GTSRB), the improvements by MePo is more limited? Why and can this be improved?

**Limitations:**

At least a few limitations of this work should be discussed in the Discussion section. Please see the identified weaknesses as a potential starting point point for the limitations discussion.

**Strengths And Weaknesses:**

Strengths:

1) (Significance) The paper addresses the most practical and general CL setting, which covers challenges like online data streams and blurry task boundaries
2) (Originality) The post-refinement strategy based on meta-learning on pseudo task sequences constructed from pretraining data is a novel contribution in the context of PTMs. Addressing the self-supervised PTM gap is a valuable and underexplored contribution.
3) (Originality) The global second-order anchoring is a concrete step forward, compared to previous work that used only first-order statistics for output alignment or per-class covariance
5) (Soundness) Computational cost experiments align with and support the targeting of a practical, real-world CL setting (GCL)
6) The paper is well-structured and written.

Weaknesses:
1) (Originality) Pseudo task sequence construction together with meta-learning to prepare models for downstream CL tasks is a well-known paradigm. Offline bi-level optimization over pseudo incremental sessions using labeled data has already been proposed in both CL and non-CL (downstream fine-tuning) contexts. E.g. MetaGCD (Wu et al., ICCV 2023) also uses bi-level meta-learning over pseudo sequential tasks from pretraining data. BiSSL (Zakarias, 2024) also uses bi-level optimization for PTM refinement in a downstream fine-tuning context.
2) (Soundness) The paper does not analyze failure cases or when MePo does not improve (e.g., DualPrompt degradation, see Q3), nor does it provide ablations on pseudo task construction hyperparameters
3) (Soundness) Claims about self-supervised PTM benefits should be validated across more backbone types (e.g., MAE-pretrained ViT, DINO, CLIP)
4) (Presentation) The connection to reconstructive memory theory is motivational rather than mechanistic. The paper does not show how the meta-covariance alignment formally implements the biological principle beyond analogy.

---

> ### Author Rebuttal · Authors · 2026-03-30
>
> ## Response to Reviewer MnQA
>
> We appreciate your constructive feedback and address your concerns point-by-point below, with all updates reserved for the final manuscript.
>
> **W1 & Q1: Difference in bi-level formulation**
>
> Compared to existing work on bi-level optimization, our method is explicitly tailored to the GCL challenges. Specifically, our inner loop simulates **single-pass, sequential online updates** to mimic catastrophic forgetting, while the outer loop enforces joint-set validation to ensure stability. This differs fundamentally from MetaGCD or BiSSL, which generally focuses on offline clustering or standard downstream fine-tuning rather than the stability-plasticity dilemma under streaming data with blurry task boundaries.
>
> Furthermore, to demonstrate that our bi-level formulation does not strictly rely on labeled data for pseudo-task construction, we evaluated a self-supervised variant of MePo. By leveraging the inherent representation capabilities of the pretrained model to cluster the data and construct pseudo-continual learning tasks, we successfully applied our meta post-refinement without requiring ground-truth labels. As shown in the table below, this self-supervised variant of MePo remains highly effective, achieving performance that is remarkably close to the supervised results.
>
> ||CIFAR-100||ImageNet-R||CUB-200||
> |-|-|-|-|-|-|-|
> ||$A_{Auc}$|$A_{Last}$|$A_{Auc}$|$A_{Last}$|$A_{Auc}$|$A_{Last}$|
> |DualPrompt| 66.36±4.42|58.09±4.40|38.63±2.19|30.71±0.82|55.73±2.77|47.08±4.94|
> |+ MePo (sup)|71.37±4.07|66.48±2.82|44.65±2.09|36.76±1.21|58.36±2.59|52.16±3.74|
> |+ MePo (self-sup)|71.26±4.07|68.11±3.08|44.35±2.03|36.40±1.36|57.91±2.62|51.48±4.10|
> |MISA|80.35±2.39|80.75±1.24|51.52±2.09|45.08±1.43|65.40±3.01|60.20±1.82|
> |+ MePo (sup)|82.30±2.83|83.99±1.35|54.86±2.20|49.18±1.38|68.13±3.17|64.75±1.00|
> |+ MePo (self-sup)|82.22±2.79|83.80±1.21|54.94±2.07|49.20±1.33|67.95±3.16|64.59±1.14|
>
> **Q2: Why MePo did not improve MVP as much as MISA**
>
> MVP relies on a contrastive loss for prompt tuning, which heavily pulls representations based on batch statistics. This dynamic contrastive pushing can sometimes conflict with the strict geometric anchoring enforced by our Meta-Cov matrix. MISA employs a simpler, non-parametric masking approach, which harmonizes more naturally with our stabilized representation space, leading to larger performance gains.
>
> **W2 & Q3: DualPrompt degradation under Sup-21/1K**
>
> DualPrompt relies on an explicit key-query matching mechanism, using distance metrics in the feature space to route input instances to specific task prompts. When MePo meta-refines the backbone (especially for models like Sup-21/1K, where the representation space undergoes significant structural realignment), the underlying feature geometry is fundamentally altered to form tighter, more separable clusters. Because DualPrompt's default key initialization and matching hyperparameters are explicitly tuned for the original, unrefined pretrained space, this spatial shift creates a geometric mismatch. The learnable prompt keys struggle to accurately anchor themselves to these newly restructured feature clusters, leading to suboptimal task routing. Consequently, this mismatch causes slight performance degradations in certain specific settings.
>
> **W3: Generality on self-supervised PTMs**
>
> Following your suggestion, we have conducted additional experiments to validate the generality of MePo in DINO architecture. We utilized 400 samples per class and trained the model for 100 meta-epochs. As shown in the table below, MePo consistently enhances the downstream GCL performance of the MISA baseline across all evaluated datasets.
>
> ||CIFAR-100||ImageNet-R||CUB-200||
> |-|-|-|-|-|-|-|
> ||$A_{Auc}$|$A_{Last}$|$A_{Auc}$|$A_{Last}$|$A_{Auc}$|$A_{Last}$|
> |MISA|52.00±3.24|54.86±4.71|42.19±0.92|37.76±1.51|27.14±3.38|33.10±4.44|
> |+ MePo|**53.54±3.31**|**56.12±5.13**|**43.53±0.99**|**38.13±1.41**|**28.24±2.76**|**34.17±4.74**|
>
> **W4: Biological inspiration**
>
> We will refined our text to clarify that reconstructive memory serves as a biological inspiration for our covariance alignment, rather than a strict mechanistic equivalent.
>
> **Q4: Limited improvements on far domains (NCH, GTSRB)**
>
> When the downstream domain (e.g., X-ray images) is drastically different from the pretraining domain (ImageNet), the reference geometry ($\Sigma_{pre}$) established during post-refinement is less representative of the downstream feature distribution. This limits the efficacy of the alignment. To improve this, MePo's Phase 1 could be conducted on a diverse, domain-generalized dataset, or a small sample of domain-specific unlabelled data could be used to initialize the Meta-Cov matrix.
>
> **Limitations statemate**
>
> We will revise our limitations to note our reliance on specific pretraining data and image tasks, future expansions to other modalities, and the lack of negative societal impacts.

---

> > ### Author Rebuttal · Reviewer_MnQA · 2026-04-02
> >
> > I appreciate the rebuttal. The self-supervised MePo variant is a meaningful addition that partially addresses W1, while "single-pass, sequential online updates" in the inner loop, is not a novel idea. Two concerns remain: (1) W3 is only partially addressed: DINO results from a single baseline, with small, potentially within-variance gains, do not convincingly establish generality; (2) The DualPrompt degradation (Q3) is explained but not resolved: if MePo is truly plug-and-play, the authors should either show how to adapt it for key-query methods or explicitly restrict the plug-in claim in the paper. I maintain my score of 4, contingent on the final manuscript addressing these in the camera-ready."

---

> > > ### Author Response · Authors · 2026-04-04
> > >
> > > We sincerely thank you for engaging in the discussion and for your constructive follow-up. We appreciate your recognition of the self-supervised MePo variant.
> > >
> > > **Concern 1 (W3 Follow-up):**
> > >
> > > Regarding the marginal DINO gains, our preliminary results relied on a default combination weight ($\alpha = 0.5$) due to strict time constraints. As shown below, properly tuning this parameter to $\alpha = 0.7$ yields substantial improvements across all benchmarks that comfortably exceed the baseline variance.
> > >
> > > |Method|&/alpha&|CIFAR-100 ($A_{Auc}$)|CIFAR-100 ($A_{Last}$)|ImageNet-R ($A_{Auc}$)|ImageNet-R ($A_{Last}$)|CUB-200 ($A_{Auc}$)|CUB-200 ($A_{Last}$)|
> > > |--|--|--|--|--|--|--|--|
> > > |MISA| - |52.00±3.24 | 54.86±4.71 | 42.19±0.92 | 37.76±1.51 | 27.14±3.38 | 33.10±4.44 |
> > > |MISA + MePo (Ours)| 0.3|52.44±3.69|55.51±4.48|42.66±1.06|37.62±1.73|27.39±2.89|33.33±5.07|
> > > |MISA + MePo (Ours)| 0.5 | 53.54±3.31 | 56.12±5.13 | 43.53±0.99 | 38.13±1.41 | 28.24±2.76 | 34.17±4.74 |
> > > |MISA + MePo (Ours)| 0.7 | **55.51±3.39**|**57.59±5.09**|**45.91±1.01**|**39.59±1.12**|**30.15±3.13**|**34.36±4.52**|
> > >
> > >
> > > **Concern 2 (Q3 Follow-up):**
> > >
> > > To avoid overclaiming, we will explicitly restrict our 'plug-and-play' claim in the final manuscript and add a dedicated paragraph in the limitations section discussing the geometric mismatch for rigid key-query methods like DualPrompt."

---

### Official Review · Reviewer_i3Y3 · 2026-03-15

**Soundness:** 2
**Presentation:** 2
**Significance:** 2
**Originality:** 2
**Overall Recommendation:** 4
**Confidence:** 2

**Summary:**

The paper tackles "General Continual Learning", in particular online learning (one epoch) tasks sequentially, as well as introducing blurry task boundaries. The paper proposes a 3-step algorithm that adapts a backbone and head, prepared for this setting by training it via (approximate) second-order meta learning. The sequential dataset observed during pretraining is created "synthetically" by mimicking the sequential tasks observed later on.

**Compliance With Llm Reviewing Policy:**

Affirmed.

**Final Justification:**

Authors addressed my concerns, although I find this kind of paper difficult to judge if you are not a researcher in this exact field - the algorithm is complicated and therefore hard to ablate and compare to related work. Nevertheless, I think this is solid work.

**Key Questions For Authors:**

Thank you for all the work put into analysing the algorithm! Remaining questions

1) I honestly dont see how Figure 6 can show that "clearly more separable than the pre-aligned features, consistent with the
significant improvements observed in Table 3 (i.e., Meta
Rep with or without Meta Cov)"

2) Also, I do not understand Figure 7.
3) I assume that step 1 is the same as in related work, i.e., how you build the pre-training stage. Generally, I don't understand the differences between related work and yours (e.g., Deep L2P, DualPrompt) that your algorithm improves upon. I guess you describe this in the appendix or so, but it's hard to grasp from the main text how this is done.
If step 1 is something you design, I would have a few questions about how this is done and ablated.
4) I honestly don't understand Fig 1. Only when reading on, I understood that you are changing parameters in Phase 2, as there are no "model updates" here anymore?
5) Sorry if this is included in the related work, but are there "naive baselines" included, such as storing a dataset as large as \theta* (plus some overhead as you save so much compute as you skip Phase 1) and train / rehearse on this in some naive way in Phase 2. There were various papers showing that these methods are very competitive a few years ago.

**Limitations:**

yes

**Strengths And Weaknesses:**

I think the authors wrote a quite detailed paper incorporating and evaluating related work to good amount. Unfortunately, I did not follow the continual learning field in the last few years, especially the important related work for this work on using large pre-trained backbones that are then adapted by "PET". I think simply put, they perform a MAML version on a continual learning setting (?). From that perspective, the algorithm does not seem particularly novel but applied to a new domain, which can be regarded a strength and a weakness, for me rather a strength as I was able to grasp the algorithm rather quickly, as it builds on very established methods in the field.

"Strengths"
The paper shows that their algorithm, which can be applied in conjuction to existing multiple related works, improves performance significantly on a variety of benchmarks. The work analyzes the sensitivity of the hyperparameter \alpha in the proposed algorithm and shows robust performance, as well as using Meta Rep / Meta Cov in Table 3. Furthermore, the authors claim that the method only endures a small amount of computational overhead (Table 2).

"Weakness"
I believe the paper proposes a complicated multi-stage algorithm with many moving bits and pieces. This, as the authors also do to some extent,  needs to be followed by a large suite of ablations allowing the reader to follow the design decision of the authors. See questions below.
Also, Table 3 seems very noisy.

---

> ### Author Rebuttal · Authors · 2026-03-30
>
> ## Response to Reviewer i3Y3
>
> We sincerely thank you for your constructive feedback. Below, we provide response to your questions, and will incorporate all additions in the final version.
>
> **Q1: Clarification on Fig. 6**
>
> While the pre-aligned (finetuned) features are separable, they are severely imbalanced due to the noisy and small online batches in general continual learning (GCL). Conversely, post-aligned features mapped to the pretrained space are comparably balanced but more crowded. By combining them via Eq. (11), MePo retains class separability of the pre-aligned features while rectifying the geometric distortion and imbalance. Fig. 6b explicitly demonstrates that the combined features form tighter clusters compared to the highly skewed pre-aligned features.
>
> **Q2: Understanding Fig. 7**
>
> Fig. 7 visualizes the activation maps of class-wise representations, where MePo achieves significantly sparser activations than strong baseline methods. Sparser activations (indicated by the darker blue areas with fewer red peaks) are highly desirable in continual learning (CL). They demonstrate that representations for different classes tend to be less overlapped, a property shown in prior CL studies [1-2] that inherently reduce catastrophic forgetting when new classes are introduced.
>
> [1] Continual semantic segmentation via repulsion-attraction of sparse and disentangled latent representations, ICCV, 2021
>
> [2] Online task-free continual learning with dynamic sparse distributed memory, ECCV, 2022
>
> **Q3: Differences from related work and Step 1**
>
> Previous work that employs pretrained models (PTMs) for CL and GCL typically focuses its optimization exclusively on the downstream CL/GCL phase. In comparison, our approach equips the upstream pretraining with an additional post-refinement using pretraining data, which is essentially an **extended pretraining phase** tailored to CL/GCL. The key idea lies in pre-adjusting pretrained representations for accommodating dynamic data distributions. To our knowledge, this is the first attempt to **prepare PTMs for CL/GCL in advance**, fundamentally different from exisiting PTMs-based CL/GCL methods.
>
> To further demonstrate the robustness and practicality of MePo, we conducted extensive ablation studies on the key hyperparameters during the offline meta-refinement phase. The results indicate that MePo is remarkably robust, achieving significant performance gains over the baselines without requiring heavy hyperparameter tuning.
>
> As shown in the Table 9 and Table 10, we have validated MePo is insensitive of **Number of pseudo-tasks** and
> **Source of pretraining data**.
>
> **Learning rate**: The meta-refinement process exhibits stable convergence across a wide range of learning rates. While a higher learning rate yields slightly better metrics, all tested rates from 0.05 to 0.5 safely and consistently outperform the unrefined baseline.
>
> MISA on Sup-21K/ImageNet-R
> |Learning rate|w/ ours|0.05|0.1|0.5|
> |-|-|-|-|-|
> |$A_{Auc}$|51.52±2.09 |54.65±2.28|54.86±2.20|55.56±2.00|
> |$A_{Last}$|45.08±1.43|48.89±1.47|49.18±1.38|49.50±0.75|
>
> **Data efficiency**: MePo is exceptionally data-efficient. Using as few as 50 samples per class matches the performance of using 200 samples and vastly outperforms the unrefined baseline. Increasing the data budget yields only marginal diminishing returns, proving that MePo does not rely on massive data availability.
>
> |Samples per class|w/ ours|50|100|200|
> |-|-|-|-|-|
> |$A_{Auc}$|51.52±2.09|54.48±2.29|54.55±2.26|54.86±2.20|
> |$A_{Last}$|45.08±1.43|48.67±1.50|48.82±1.53|49.18±1.38|
>
> **Q4: Clarification on Fig. 1**
>
> In Phase 2 (the downstream GCL phase), the meta-refined backbone ($\theta^*$) is strictly frozen. The "model updates" in this phase only apply to the lightweight parameter-efficient tuning (PET) modules (e.g., prompts) and the output head, mimicking the standard rehearsal-free PTM setting.
>
> **Q5: Naive replay baselines**
>
> We mainly target a **rehearsal-free** setting, since storing a large pretraining dataset for downstream rehearsal may lead to significant resource overheads and potential privacy issues.
> Following your suggestion, we additionally evaluated MePo in a **rehearsal-based** setting. We augmented the MISA [2] with memory replay buffers of 500 and 2000 samples, using the Sup-21K backbone on the ImageNet-R benchmark. As shown in the table below, MePo remains highly effective even when memory replay is explicitly introduced. This demonstrates that the structural knowledge condensed during MePo's offline refinement provides orthogonal benefits that are highly complementary to standard rehearsal strategies.
>
> |Buffer size|Method|$A_auc$|$A_last$|
> |-|-|-|-|
> |500|MISA[1] | 56.35±1.05| 49.67±0.29|
> |-| w/ ours|**60.19±1.13**| **54.88±0.36**|
> |2000|MISA[1] |60.03±0.86|56.49±0.66|
> |-|w/ ours|**63.41±1.01**|**60.91±0.38**|
>
> [1] Advancing prompt-based methods for replay-independent general continual learning, ICLR, 2025

---

> > ### Author Rebuttal · Reviewer_i3Y3 · 2026-04-04
> >
> > Thanks for the rebuttal! Final reservations remain, I will keep my score.

---

> > > ### Author Response · Authors · 2026-04-04
> > >
> > > We sincerely thank you for your time and valuable feedback. Please do not hesitate to let us know if you have any further questions.

---

### Decision · Program_Chairs · 2026-04-30

**Decision:**

Accept (regular)

**Comment:**

Reviewers agree that this paper is tackling a hard problem (General Continual Learning), and is well-motivated and well-written. They liked the idea of meta-training on a pre-trained model, as well as using second order information through a covariance matrix. There were questions about relating this to existing meta-continual learning work (which I thought the authors rebutted sufficiently), sensitivity to some choices such as pseudo-tasks (which the authors rebutted well), questions about when the method fails or works well, and some additional simpler baselines. The authors rebutted these points. The idea itself is an application of meta-learning ideas to GCL but for pre-trained models, and I recommend acceptance.